# TDP-43 forms amyloid filaments with a distinct fold in type A FTLD-TDP

Diana Arseni[1], Renren Chen[1], Alexey G. Murzin[1], Sew Y. Peak-Chew[1], Holly J. Garringer[2], Kathy L. Newell[2], Fuyuki Kametani[3], Andrew C. Robinson[4], Ruben Vidal[2], Bernardino Ghetti[2], Masato Hasegawa[3] & Benjamin Ryskeldi-Falcon[1✉]

The abnormal assembly of TAR DNA-binding protein 43 (TDP-43) in neuronal and glial cells characterizes nearly all cases of amyotrophic lateral sclerosis (ALS) and around half of cases of frontotemporal lobar degeneration (FTLD)[1,2]. A causal role for TDP-43 assembly in neurodegeneration is evidenced by dominantly inherited missense mutations in *TARDBP*, the gene encoding TDP-43, that promote assembly and give rise to ALS and FTLD[3–7]. At least four types (A–D) of FTLD with TDP-43 pathology (FTLD-TDP) are defined by distinct brain distributions of assembled TDP-43 and are associated with different clinical presentations of frontotemporal dementia[8]. We previously showed, using cryo-electron microscopy, that TDP-43 assembles into amyloid filaments in ALS and type B FTLD-TDP[9]. However, the structures of assembled TDP-43 in FTLD without ALS remained unknown. Here we report the cryo-electron microscopy structures of assembled TDP-43 from the brains of three individuals with the most common type of FTLD-TDP, type A. TDP-43 formed amyloid filaments with a new fold that was the same across individuals, indicating that this fold may characterize type A FTLD-TDP. The fold resembles a chevron badge and is unlike the double-spiral-shaped fold of ALS and type B FTLD-TDP, establishing that distinct filament folds of TDP-43 characterize different neurodegenerative conditions. The structures, in combination with mass spectrometry, led to the identification of two new post-translational modifications of assembled TDP-43, citrullination and monomethylation of R293, and indicate that they may facilitate filament formation and observed structural variation in individual filaments. The structures of TDP-43 filaments from type A FTLD-TDP will guide mechanistic studies of TDP-43 assembly, as well as the development of diagnostic and therapeutic compounds for TDP-43 proteinopathies.

TAR DNA-binding protein 43 (TDP-43) is a ubiquitously expressed RNA-binding protein with diverse roles in RNA processing. It mainly resides in nuclear ribonucleoprotein granules but also undergoes nucleocytoplasmic shuttling to participate in cytoplasmic ribonucleoprotein granules[10]. The amino-terminal part of TDP-43 includes a DIX (dishevelled and axin) domain, a nuclear localization signal, and tandem RNA-recognition motifs (RRMs). The carboxy-terminal part comprises an intrinsically disordered low-complexity domain (LCD), which contains regions enriched in glycine, hydrophobic residues, and glutamine and asparagine (Q/N). The DIX domain, RRMs and LCD all contribute to the association of TDP-43 with ribonucleoprotein granules and RNA[10,11].

In disease, full-length TDP-43 and abnormally truncated C-terminal fragments (CTFs) assemble and are ubiquitylated and phosphorylated[1,2]. The assemblies have granulofilamentous morphologies, with diameters of 10–15 nm (refs. 9,12–19). They bind the amyloidophilic dye thioflavin-S poorly[20]. Cryo-electron microscopy (cryo-EM) structures of assembled TDP-43 from the prefrontal and motor cortices of two individuals with amyotrophic lateral sclerosis (ALS) and type B frontotemporal lobar degeneration with TDP-43 pathology (FTLD-TDP) have shown amyloid filaments with an identical double-spiral-shaped fold (double-spiral fold)[9]. The ordered core of these filaments is formed by the N-terminal half of the LCD (G282–Q360), with the flanking regions forming a fuzzy coat.

The structures of assembled TDP-43 in other neurodegenerative conditions were unknown. A recent cryo-EM study involving four individuals with FTLD-TDP types A–D in the absence of ALS did not find amyloid filaments of TDP-43 and reported that filaments of transmembrane protein 106B (TMEM106B) characterize FTLD-TDP instead[21]. However, other studies have shown that TMEM106B filaments accumulate in the human brain in an age-dependent manner in many neurodegenerative conditions, including tauopathies and synucleinopathies, as well as in neurologically normal individuals[22–25].

Here, we use cryo-EM to determine the structures of assembled TDP-43 from the brains of individuals with the most common type

[1]MRC Laboratory of Molecular Biology, Cambridge, UK. [2]Department of Pathology and Laboratory Medicine, Indiana University School of Medicine, Indianapolis, IN, USA. [3]Department of Brain and Neurosciences, Tokyo Metropolitan Institute of Medical Science, Tokyo, Japan. [4]Division of Neuroscience, Faculty of Biology, Medicine and Health, School of Biological Sciences, University of Manchester, Salford Royal Hospital, Salford, UK. ✉e-mail: bfalcon@mrc-lmb.cam.ac.uk

of FTLD-TDP, type A. We show that TDP-43 does in fact form amyloid filaments, which are present in addition to TMEM106B filaments. The structures reveal that unlike TMEM106B, TDP-43 forms distinct amyloid filament folds in different neurodegenerative conditions and detail their structural basis.

## TDP-43 filaments in type A FTLD-TDP

We extracted assembled TDP-43 from the prefrontal cortex of individuals with type A FTLD-TDP (Extended Data Table 1), using the method we developed for ALS with type B FTLD-TDP[9]. Four individuals carried mutations in *GRN* associated with type A FTLD-TDP[8,26,27], whereas one individual had wild-type *GRN*. All individuals had abundant compact neuronal cytoplasmic inclusions, short dystrophic neurites and infrequent neuronal intranuclear inclusions of assembled TDP-43 concentrated in the second and third cortical layers, indicative of type A FTLD-TDP[8] (Extended Data Fig. 1a,b). Immunoblotting of the extracts showed that the assemblies were composed of both full-length TDP-43 and CTFs and were phosphorylated at S409 and S410, as previously observed[1,2] (Extended Data Fig. 1c).

We used cryo-EM to image the extracts from the prefrontal cortex of two of the individuals with *GRN* mutations and the individual with wild-type *GRN* (Extended Data Fig. 1d). This revealed a population of straight, unbranched filaments with granular surfaces and projected widths of approximately 10–15 nm, consistent with previous reports of TDP-43 filaments in situ in the brains of individuals with FTLD-TDP[13,15], as well as in brain extracts[9,17–19]. The TDP-43 identity of the filaments was confirmed using immunogold negative-stain electron microscopy (Extended Data Fig. 1e).

We determined the structures of the ordered cores of the TDP-43 filaments for each of the three individuals independently using helical reconstruction of the cryo-EM images, with resolutions of up to 2.4 Å (Fig. 1a, Extended Data Fig. 2a–c and Extended Data Table 2). The filaments comprised a single amyloid protofilament of stacked TDP-43 molecules. The reconstructions revealed a new TDP-43 filament fold that was identical among individuals, irrespective of genetic variation in *GRN*. Our results indicate that this TDP-43 amyloid filament fold may characterize type A FTLD-TDP.

## TMEM106B filaments in type A FTLD-TDP

Single and double TMEM106B filaments were also observed in the cryo-EM images for each individual, on the basis of their characteristic projected widths of 12 and 26 nm, respectively, their blunt ends and their lack of surface granularity (Extended Data Fig. 3a), as previously reported in FTLD-TDP[21–23]. The presence of TMEM106B was confirmed using mass spectrometry, which identified peptides mapping to the C-terminal region that forms the ordered core of the filaments[21–23] (Extended Data Fig. 3b and Supplementary Table 1).

The presence of both TDP-43 filaments and TMEM106B filaments in type A FTLD-TDP, as well as in ALS with type B FTLD-TDP[9,22], is at odds with a recent report of amyloid filaments in FTLD-TDP being composed of TMEM106B but not TDP-43 (ref. 21). Differences in brain extraction protocols may account for this discrepancy. We found TDP-43 filaments in the supernatant following centrifugation at 27,000*g*, whereas the other study examined the pellet following centrifugation at 21,000*g*. TMEM106B filaments have also been observed in the brains of individuals with tauopathies and α-synucleinopathies, as well as in the brains of neurologically normal individuals[22–25]. TMEM106B inclusions do not colocalize with inclusions of TDP-43, tau or α-synuclein[28]. In contrast to these proteins, there are no clear relationships between different TMEM106B filament folds and diseases. Available evidence is consistent with the age-dependent accumulation of TMEM106B filaments in the human brain[22,28], which may be modified by the *TMEM106B* haplotype[29].

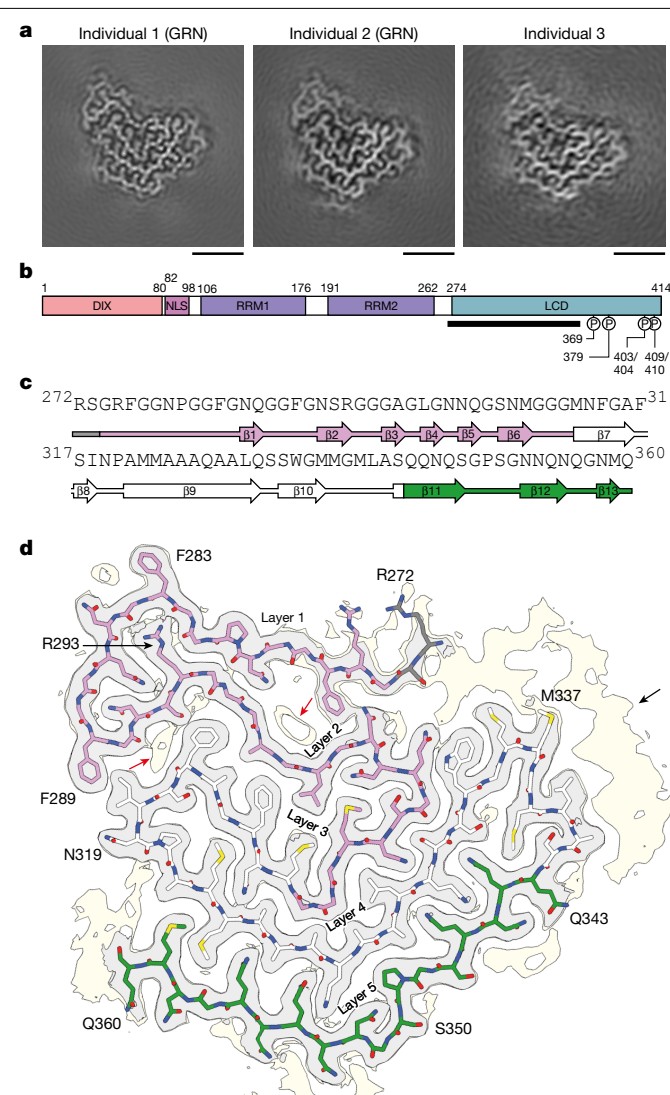

**Fig. 1 | Cryo-EM structures of TDP-43 amyloid filaments from individuals with type A FTLD-TDP. a**, Cryo-EM maps of TDP-43 filaments from the prefrontal cortex of three individuals with type A FTLD-TDP, shown as central slices perpendicular to the helical axis. GRN indicates individuals with mutations in *GRN* associated with type A FTLD-TDP. Scale bars, 25 Å. **b**, Schematic of the domain organization of TDP-43. Disease-associated phosphorylation sites are shown. The black line indicates the region that forms the ordered filament fold. NLS, nuclear localization signal. **c**, Amino acid sequence alignment of the secondary structure elements of the filament fold. Arrows indicate β-strands. **d**, Cryo-EM map shown at high (grey) and low (yellow) contour levels and atomic model, shown for a single TDP-43 molecule perpendicular to the helical axis. The five layers of the filament fold are labelled. Less well-resolved protein-like density extending from R272 (black arrow), an isolated peptide-like density adjacent to G351–N355 (yellow arrow), and non-protein densities within cavities between layers 1 to 3 (red arrows) are indicated. In **c** and **d**, the glycine-rich (G274–G310, magenta), hydrophobic (M311–S342, white) and Q/N-rich (Q343–Q360, green) regions are highlighted.

## TDP-43 filament fold of type A FTLD-TDP

Our cryo-EM reconstructions were of sufficient resolution to visualize peptide groups and ordered solvent, enabling us to build an accurate atomic model of the TDP-43 filament fold and establish its right-handed helical twist (Fig. 1b–d and Extended Data Figs. 2d,e and 4a,b). The fold consists of five connected layers formed by R272–Q360 in

the LCD. The first two layers are formed by the glycine-rich region (G274–G310), the hydrophobic region (M311–S342) contributes the third and fourth layers, and the Q/N-rich region (Q343–Q360) forms the fifth layer (Extended Data Fig. 4c,d).

The fold is centred around a kinked, 11-residue β-strand formed by A321–Q331 in the fourth layer, which forms steric zippers with β-strands in the neighbouring third and fifth layers and imparts an arrangement that resembles a three-bar chevron badge (Extended Data Fig. 4e). Owing to this resemblance, we hereafter refer to the fold as the chevron fold. The first and second layers contain only short, 2- to 3-residue β-strands that do not participate in zipper packing. In addition to hydrogen bonding within intermolecular β-sheets, the filaments are stabilized by hydrogen bonding ladders formed by glutamine and asparagine side chains and by staggered stacking interactions between aromatic residues. The layers associate through hydrogen bonds between abundant neutral polar residues and glycine (Extended Data Fig. 4f). The second and third layers also interact through a cluster of hydrophobic residues between A297 and F316 (Extended Data Fig. 4g). Three arginine residues at the N terminus comprise the only charged residues in the fold. Two of these, R272 and R275, are in the solvent-exposed first layer. The third, R293, is located in an unusual buried position inside a compact loop linking the first and second layers.

Less-ordered protein density extends from the N-terminal residue R272 and covers a hydrophobic patch formed by W334–A341 in the turn linking the fourth and fifth layers (Fig. 1c,d). This density could accommodate approximately 17 residues (V255–E271), which would include β5 of the second TDP-43 RRM. Therefore, residues N-terminal to V255 and C-terminal to Q360 form the fuzzy coat of the filaments. The presence of CTFs that lack all or some of V255–E271 in the filaments may explain why this extra protein density was less ordered[30]. Their presence would lead to the exposure of the hydrophobic patch (W334–A341). Hydrophobic surface patches on amyloid filaments are thought to engage in aberrant interactions[31].

Another peptide-like density is located on the outside surface of the fifth layer, adjacent to G351–N355 (Fig. 1a,d). Its disconnected nature precluded sequence assignment. It may originate from the N-terminal or C-terminal flanking regions or from a separate interacting protein. Disconnected peptides have previously been shown to associate with α-synuclein filaments[32]. We also observed non-proteinaceous densities within cavities between the first, second and third layers (Fig. 1d). Unlike the protein densities, they appeared partly contiguous along the helical axis (Extended Data Fig. 4h), possibly because they do not follow the same helical symmetry as TDP-43. Buried non-proteinaceous densities have also been observed in tau and α-synuclein filaments from human brain and may represent cofactors for filament formation[33,34].

## Structural variation of TDP-43 filaments

Three-dimensional (3D) classification of the cryo-EM filament segments from the individual with the largest dataset revealed alternative conformations of the N-terminal region R272–G295 and of the turn G335–Q343 linking the fourth and fifth layers (Fig. 2a–c and Extended Data Fig. 5a–d). Fewer than 5% of the filament segments contributed to classes with these alternative conformations, demonstrating that they were rare (Extended Data Table 2). No variation was observed at regions of zipper packing between β-strands. The alternative conformation of the N-terminal region, but not of the turn linking the fourth and fifth layers, was observed for the second individual, and no alternative conformations were observed for the third individual, consistent with the smaller sizes of their datasets (Extended Data Table 2).

We determined the structures of filament segments containing the alternative conformations, with resolutions of up to 2.5 Å (Fig. 2a and Extended Data Fig. 5). In the alternative conformation of the N-terminal region, residues G281–G295 follow approximately the same path as residues R272–P280 in the main conformation, exposing R293 on

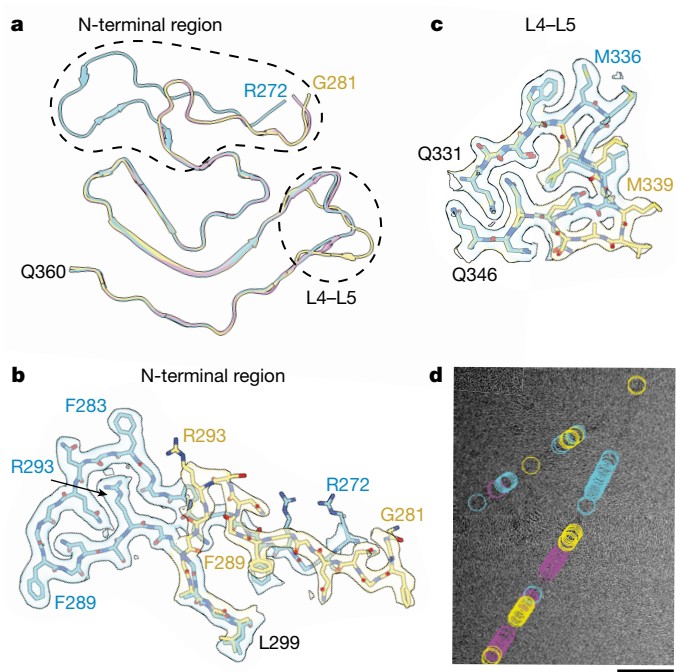

**Fig. 2 | Alternative local conformations of TDP-43 amyloid filaments from type A FTLD-TDP. a**, Overlay of atomic models of TDP-43 filaments from type A FTLD-TDP with different local conformations of the N-terminal region and of the turn connecting the fourth layer to the fifth (L4–L5). **b,c**, Cryo-EM maps and atomic models with different local conformations of the the N-terminal region (**b**) and of the turn connecting the fourth layer to the fifth (L4–L5) (**c**), shown for a single TDP-43 molecule perpendicular to the helical axis. **d**, Example micrograph showing the positions of filament segments contributing to cryo-EM maps with different local conformations, which occur in individual filaments. Scale bar, 50 nm. Further examples are shown in Extended Data Fig. 6. In **a**–**d**, the main conformation is shown in cyan, the alternative local conformation of the N-terminal region in magenta, and the alternative local conformation of the N-terminal region and the turn connecting the fourth layer to the fifth in yellow.

the filament surface, and residues R272–P280 are in the fuzzy coat (Fig. 2b). In the alternative conformation of the turn linking the fourth and fifth layers, several residues switch their interior–exterior positions, with M336, A341 and Q343 becoming buried, whereas M339 and S342 become surface exposed (Fig. 2c).

Mapping of the filament segment locations in the micrographs revealed that TDP-43 molecules with different local structural variations could coexist in individual filaments (Fig. 2d and Extended Data Fig. 6). Segments of the same variant structures grouped together into blocks of variable length, with no apparent directionality to their arrangement. Local structural variation at turn regions has also been observed in individual filaments of amyloid light-chain protein[35]. These results show that amyloid filaments do not always adopt uniform repetitive structures. The structural variability of TDP-43 amyloid filaments may, therefore, have implications for the development of diagnostic and therapeutic compounds.

## Post-translational modification of R293

The side chain of R293 is completely buried in the main conformation of the chevron fold, and its positive charge is only partially compensated by hydrogen bonds to the surrounding peptide groups (Fig. 3a). The overall effect of this buried charge is expected to be destabilizing. We proposed the hypothesis that the burial of this residue may have been facilitated by the post-translational modification citrullination (deimination), in which the charged guanidinium group of arginine

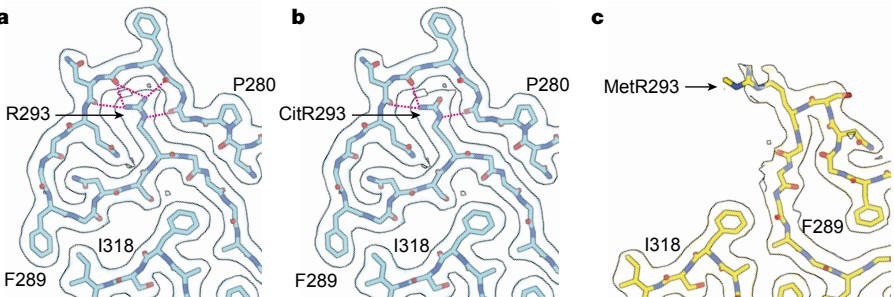

**Fig. 3 | Post-translational modifications of R293 in TDP-43 amyloid filaments from type A FTLD-TDP. a–c,** Cryo-EM maps and atomic models of TDP-43 filaments from type A FTLD-TDP around R293 with the main conformation of the N-terminal region and unmodified R293 (**a**), the main conformation of the N-terminal region and citrullinated R293 (CitR293) (**b**), and the alternative conformation of the N-terminal region and monomethylated R293 (MetR293) (**c**). The maps and models are shown for a single TDP-43 molecule perpendicular to the helical axis. Hydrogen bonds to the side chains of unmodified and modified R293 are shown as dashed magenta lines.

is hydrolysed, yielding a neutral ureido group (Fig. 3b). Analysis of filaments from the brains of the individuals with type A FTLD-TDP by mass spectrometry established the presence of TDP-43 molecules that were citrullinated at R293 (Extended Data Fig. 7). Our results indicate that citrullination may facilitate the formation of TDP-43 filaments in type A FTLD-TDP by removing the charge of R293.

Our mass spectrometry analysis also revealed that R293 was monomethylated in some TDP-43 molecules (Extended Data Fig. 8). Monomethylation of R293 has also previously been observed under physiological conditions[36]. Methyl-R293 is incompatible with the main conformation of the filament fold, owing to steric clashes, and can only be accommodated in the alternative conformation of the N-terminal region, where its side chain is exposed (Fig. 3c). As this alternative conformation is rare, R293 can only be methylated in a minority of TDP-43 molecules. Monomethylation of R293 has been reported to reduce TDP-43 assembly in vitro[37].

Arginine 293 is located in the sole RGG/RG motif of the TDP-43 LCD. RGG/RG motifs participate in functional interactions of RNA-binding proteins and are regulated by arginine citrullination and methylation[38].

Peptidyl-arginine deiminase 4 (PAD4) can citrullinate arginine residues in RGG motifs[39], and several peptidyl-arginine methyltransferases can target this motif[38]. Increased PAD activity and protein citrullination have been observed in several neurodegenerative diseases[40]. Possible roles for the citrullination and methylation of TDP-43 at R293 in other neurodegenerative conditions, as well as in healthy individuals, remain to be determined. Together, our results indicate that post-translational modifications of R293 may influence local structural variation of TDP-43 filaments from type A FTLD-TDP by altering the charge and size of the residue.

## Comparison of TDP-43 filament folds

The chevron fold of type A FTLD-TDP is unlike the double-spiral fold of ALS with type B FTLD-TDP, demonstrating that distinct amyloid filament folds of TDP-43 characterize different neurodegenerative conditions (Fig. 4). This supports the broader notion that distinct amyloid filament folds of specific proteins underlie neurodegenerative disease[32,41].

The chevron fold of type A FTLD-TDP is formed by the same part of the TDP-43 LCD as the double-spiral fold of ALS and type B FTLD-TDP,

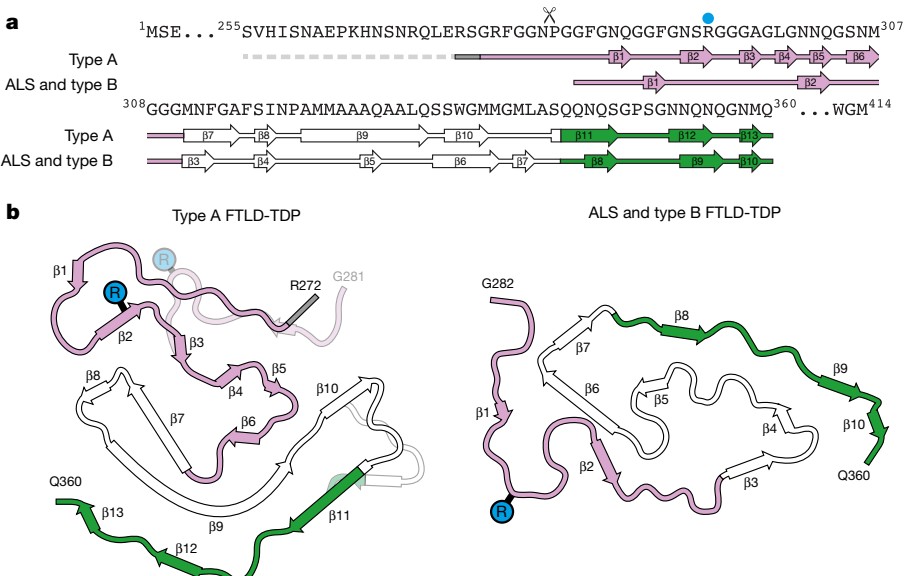

**Fig. 4 | Comparison of the TDP-43 amyloid filament folds of type A FTLD-TDP and of ALS and type B FTLD-TDP. a,** Amino acid sequence alignment of the secondary structure elements of the TDP-43 filament folds of type A FTLD-TDP and of ALS and type B FTLD-TDP (PDB 7PY2). The N-terminal truncation site at P280 is indicated by a scissor symbol. R293 is indicated by a blue dot. **b,** Schematic of the secondary structure elements of the filament folds, shown for a single TDP-43 molecule perpendicular to the helical axis. Alternative local conformations of the type A FTLD-TDP filament fold are transparent, and R293 is highlighted. In **a** and **b**, arrows indicate β-strands. The glycine-rich (G274–G310, magenta), hydrophobic (M311–S342, white) and Q/N-rich (Q343–Q360, green) regions are highlighted.

plus ten further residues at the N terminus (R272–G281). These further residues may account for the presence of distinct CTFs in type A FTLD-TDP compared with ALS and type B FTLD-TDP[42]. A cellular model of TDP-43 assembly indicated that N-terminal truncation may occur after filament formation[17]. A CTF beginning at P280 has been found in ALS[30] but could not be generated from the main conformation of the type A FTLD-TDP filament fold.

Residues A321–Q331 in the hydrophobic region form the nexus of both filament folds. These residues are conserved and may form cooperative α-helices and intermolecular β-sheets of functional importance[43,44]. Differences between the two filament folds arise from distinct structural organizations of the hydrophobic region. In the double-spiral fold, this region forms two compact hydrophobic clusters, whereas in the chevron fold it adopts an extended conformation owing to its more extensive β-structure, which is stabilized by zipper packing with β-strands of the glycine-rich and Q/N-rich regions.

Like the double-spiral fold, the surfaces of the chevron fold are formed of the glycine-rich and Q/N-rich regions of TDP-43 and lack charged grooves (Extended Data Fig. 4i), which possibly accounts for the poor binding of amyloid imaging ligands[20,45]. The Q/N-rich region has an extended conformation and similar secondary structure in both folds, but the interior–exterior orientations of its side chains are reversed. This includes S347 and S350, which are buried in the double-spiral fold but solvent exposed in the chevron fold. Recently, phosphorylation of S350 was identified in assembled TDP-43 from an individual with type A FTLD-TDP but not that from individuals with ALS and type B FTLD-TDP[18]. It remains to be determined whether phosphorylation occurs before or after filament formation.

Twenty-four disease-associated TARDBP mutations are located in the region that forms the filament folds of type A FTLD-TDP and ALS with type B FTLD-TDP. Most of these mutations are compatible with at least one of the filament folds, whereas four mutations are incompatible with both folds (Supplementary Data Table 2). Given the observed structural variation of the type A FTLD-TDP filament fold, we do not exclude the possibility that these mutations could be accommodated in these folds by other alternative local conformations. It is also possible that individuals with these mutations have different filament folds.

The differences between the two filament folds may account for the distinct seeding abilities and toxicities of assembled TDP-43 from type A FTLD-TDP and that from ALS and type B FTLD-TDP in cellular and animal models[17,46–48]. This may underlie the distinct pathologies of the different types of FTLD-TDP. The different types of FTLD-TDP are distinguished by the brain distribution of assembled TDP-43 (ref. 8), which indicates that the local environment may influence TDP-43 filament folds. Producing disease-associated filament folds in model systems will be key to testing these hypotheses. So far, TDP-43 filaments assembled in vitro have not recapitulated structures from brain[49–51].

## Conclusions

Here, we have established that TDP-43 forms amyloid filaments in type A FTLD-TDP, as it does in ALS with type B FTLD-TDP[9]. The chevron fold of type A FTLD-TDP is unlike the double-spiral fold of ALS and type B FTLD-TDP, demonstrating that distinct TDP-43 amyloid filament folds characterize different neurodegenerative conditions. The structures indicate a role for post-translational modifications of arginine in filament formation and in structural variation in individual filaments. This work will guide mechanistic studies of TDP-43 assembly, as well as the development of diagnostic and therapeutic compounds targeting assembled TDP-43.

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

## Methods

### Human tissue samples

Human tissue samples were from the Brain Library of the Dementia Laboratory at Indiana University School of Medicine (individuals 2, 4 and 5) and the Manchester Brain Bank (individuals 1 and 3). Their use in this study was approved by the ethical review processes at each institution. Informed consent was obtained from the patients' next of kin. The individuals were selected based on a neuropathological diagnosis of type A FTLD-TDP, according to the criteria set out in ref. 8, as well as a lack of comorbid neurodegenerative-disease-associated neuropathology. Individual 2 has been described before as patient 3 in ref. 52. All individuals had abundant compact neuronal cytoplasmic inclusions, short dystrophic neurites and infrequent neuronal intranuclear inclusions concentrated in cortical layers two and three that were detected using antibodies against the N terminus of TDP-43 and TDP-43 phosphorylated at S409 and S410. All individuals received clinical diagnoses of frontotemporal dementia. The clinical presentations of individuals 1, 2, 4 and 5 were consistent with non-fluent variant primary progressive aphasia, whereas individual 3 presented with clinical symptoms consistent with behavioural-variant frontotemporal dementia. These clinical presentations have previously been found to be associated with type A FTLD-TDP[8]. Individual 3 had wild-type *GRN*, whereas the remaining four individuals had mutations in *GRN* associated with type A FTLD-TDP. All individuals had normal *C9orf72* hexanucleotide repeat numbers. Further clinicopathological details are given in Extended Data Table 1.

### Genetic analyses

Whole-exome sequencing target enrichment made use of the SureSelectTX human all-exon library (V6, 58 megabase pairs; Agilent), and high-throughput sequencing was carried out using a HiSeq 4000 (sx75 base-pair paired-end configuration; Illumina). To screen for hexanucleotide repeat expansions in the *C9orf72* gene, we performed repeat-primed PCR followed by fragment length analyses as previously described[53]. Oligonucleotides designed to amplify the coding exons and corresponding flanking intronic regions of the *GRN* gene were used for PCR using 50 ng of genomic DNA extracted from brain tissue. The amplified products were purified and subjected to direct dideoxy sequencing as previously described[54].

### Extraction of pathological assembled TDP-43

Assembled TDP-43 was extracted from flash-frozen prefrontal cortex as previously described[9]. Grey matter was dissected from flash-frozen prefrontal cortex and homogenized using a Polytron (Kinematica) in 40 volumes (v/w) of extraction buffer containing 10 mM Tris-HCl pH 7.5, 0.8 M NaCl, 10% sucrose and 1 mM EGTA. A 25% solution of sarkosyl in water was added to the homogenates to achieve a final concentration of 2% sarkosyl. The homogenates were then incubated for 1 h at 37 °C with orbital shaking at 200 rpm, followed by centrifugation at 27,000*g* for 10 min. The supernatants were retained and centrifuged at 166,000*g* for 20 min. The pellets were resuspended in 6 ml g$^{-1}$ tissue of extraction buffer containing 1% sarkosyl by sonication for 5 min at 50% amplitude (Qsonica Q700) then diluted fourfold with the same buffer and incubated for 30 min at 37 °C with orbital shaking at 200 rpm. The samples were then centrifuged at 17,000*g* for 5 min, and the supernatants were retained and centrifuged at 166,000*g* for 20 min. The pellets were resuspended in 1 ml g$^{-1}$ tissue of extraction buffer containing 1% sarkosyl by incubation for 1 h at 37 °C with orbital shaking at 200 rpm. The samples were centrifuged at 100,000*g* for 20 min, and the pellets were resuspended in 30 μl g$^{-1}$ tissue of 20 mM Tris-HCl pH 7.4, 150 mM NaCl by sonication for 5 min at 50% amplitude (Qsonica Q700). One to two grams of tissue was used for each cryo-EM sample. All centrifugation steps were carried out at 25 °C.

### Immunolabelling

For histology, brain hemispheres were fixed with 10% buffered formalin and embedded in paraffin. Deparaffinized sections (8 μm thick) were incubated in 10 mM sodium citrate buffer at 105 °C for 10 min and treated with 95% formic acid for 5 min. After washing, sections were blocked with 10% fetal calf serum in phosphate-buffered saline (PBS) and then incubated overnight with primary antibodies against the TDP-43 N terminus (Abcam ab57105, 1:10,000, or Proteintech 10782-2-AP, 1:2,000) and pS409/410 TDP-43 (CosmoBio CAC-TIP-PTD-M01, 1:1,000) in PBS containing 10% fetal calf serum. After incubation of sections with biotinylated secondary antibodies for 2 h, labelling was detected using an ABC staining kit (Vector) with DAB. Sections were counterstained with haematoxylin.

For immunoblotting, sarkosyl-soluble and sarkosyl-insoluble brain extracts were resolved using 12% or 4–12% BIS-Tris gels (Novex) at 200 V for 45 min and transferred on to nitrocellulose membranes. Membranes were blocked in PBS containing 1% bovine serum albumin and 0.2% Tween for 30 min at 21 °C and incubated with a primary antibody against pS409/410 TDP-43 (CosmoBio CAC-TIP-PTD-M01, 1:3,000) at 21 °C for 1 h. Membranes were then washed three times with PBS containing 0.2% Tween and incubated with fluorescent StarBright Blue 520 (Bio-Rad) or biotinylated anti-mouse IgG (Vector Lab Inc) secondary antibodies. Membranes were then washed three times with PBS containing 0.2% Tween and imaged using a ChemiDoc MP (Bio-Rad).

For immunogold negative-stain electron microscopy, sarkosyl-insoluble brain extracts were deposited on to carbon-coated 300-mesh copper grids (Nissin EM), blocked with 0.1% gelatin in PBS and incubated with a primary antibody against pS409/410 TDP-43 (CosmoBio CAC-TIP-PTD-M01) at a dilution of 1:100 in 0.1% gelatin in PBS at 21 °C for 3 h. After washing with PBS, the grids were incubated with secondary antibodies conjugated to 10-nm gold particles (Cytodiagnostics) at a dilution of 1:20 in 0.1% gelatin in PBS at 21 °C for 1 h. The grids were then stained with 2% uranyl acetate. Electron micrograph images were acquired using a 120-keV Thermo Fisher Scientific Tecnai Spirit or a 80-keV JEOL JEM-1400 electron microscope equipped with charge-coupled device cameras.

### Mass spectrometry

Assembled TDP-43 extracted from 0.1 g of tissue was incubated with 0.4 mg ml$^{-1}$ pronase (Sigma) for 1 h at 21 °C and was pelleted by centrifugation at 166,000*g* for 20 min. The pellet was resuspended in 100 μl hexafluoroisopropanol and incubated at 37 °C overnight. The resuspended pellets were then sonicated for 2 min at 50% amplitude (QSonica Q700), followed by incubation at 37 °C for 2 h. This sonication and incubation cycle was repeated until all material was visibly resuspended. The resuspended pellets were centrifuged at 166,000*g* for 15 min, and the supernatant was collected and dried by vacuum centrifugation (Savant). The dried protein samples were resuspended in 50 mM ammonium bicarbonate containing 8 M urea, reduced with 5 mM DTT at 56 °C for 30 min and alkylated with 10 mM iodoacetamide in the dark at room temperature for 30 min. Samples were diluted to 1 M urea with 50 mM ammonium bicarbonate and incubated with chymotrypsin (Promega) at 25 °C overnight. Protease digestion was stopped by the addition of formic acid to a final concentration of 0.5%. Samples were then centrifuged at 16,000*g* for 5 min, and the supernatants were desalted and fractionated using custom-made C18 stop-and-go-extraction (STAGE) tips (3M Empore) packed with porous oligo R3 resin (Thermo Fisher). The STAGE tips were equilibrated with 80% acetonitrile (MeCN) containing 0.5% formic acid, followed by 0.5% formic acid. Bound peptides were eluted stepwise with MeCN at concentrations increasing from 5% to 60% MeCN in 10 mM ammonium bicarbonate and partially dried down by vacuum centrifugation (Savant).

Fractionated peptides were analysed by liquid chromatography coupled with tandem mass spectrometry (LC-MS/MS) using a fully

automated Ultimate 3000 RSLCnano system (Thermo Fisher). Peptides were trapped by a 100 μm × 2 cm PepMap 100 C18 nano trap column (Thermo Scientific) and separated on a 75 μm × 25 cm nanoEase M/Z HSS C18 T3 column (Waters) using a binary gradient formed of 2% MeCN in 0.1% formic acid (buffer A) and 80% MeCN in 0.1% formic acid (buffer B) at a flow rate of 300 nl min⁻¹. Eluted peptides were introduced directly using a nanoFlex ion source into an Orbitrap Eclipse mass spectrometer (Thermo Fisher). MS1 spectra were acquired at a resolution of 120K, mass range of 380–1400 $m/z$, automatic gain control target of 4e5, maximum injection time of 50 ms and dynamic exclusion of 60 s. MS2 analysis was carried out with higher energy collisional dissociation activation orbitrap detection with a resolution of 15K, automatic gain control target of 5e4, maximum injection time of 50 ms, normalized collision energy of 30% and isolation window of 1.2 $m/z$.

LC-MS/MS data were searched against the human-reviewed database (UniProtKB/Swiss-Prot, release 2019_03) using Mascot (Matrix Science, v.2.4). Database search parameters were set with a precursor tolerance of 10 ppm and a fragment ion mass tolerance of 0.1 Da. A maximum of three missed chymotrypsin cleavages were allowed. Carbamidomethyl cysteine was set as static modification. Methionine oxidation, arginine citrullination and methylation, and asparagine and glutamine deamination were specified as variable modifications. Scaffold (v.4, Proteome Software Inc.) was used to validate MS/MS-based peptide and protein identifications. MS/MS spectra containing arginine citrullination and methylation were manually confirmed.

### Cryo-EM
Extracted assembled TDP-43 was incubated with 0.4 mg ml⁻¹ pronase (Sigma) for 1 h at 21 °C and centrifuged at 3,000$g$ for 15 s. The supernatants were retained and applied to glow-discharged 1.2/1.3 μm holey carbon-coated 300-mesh gold grids (Quantifoil) and plunge-frozen in liquid ethane using a Vitrobot Mark IV (Thermo Fisher). Images were acquired using a 300 keV Titan Krios microscope (Thermo Fisher) equipped with a K3 detector (Gatan) and a GIF Quantum energy filter (Gatan) operated at a slit width of 20 eV. Aberration-free image shift in the EPU software (Thermo Fisher) was used during image acquisition. Further details are given in Extended Data Table 2.

### Helical reconstruction
Video frames were gain-corrected, aligned, dose-weighted and summed using the motion correction program in RELION-4.0 (ref. 55). The motion-corrected micrographs were used to estimate the contrast transfer function using CTFFIND-4.1 (ref. 56). All subsequent image processing used helical reconstruction methods in RELION-4.0 (refs. 57,58). TDP-43 filaments were picked manually, and reference-free two-dimensional (2D) classification was performed to remove suboptimal segments. Initial 3D reference models were generated de novo by producing sinograms from 2D class averages as previously described[59]. Then, 3D autorefinements with optimization of the helical twist were performed, followed by Bayesian polishing and contrast transfer function refinement[55,60]; 3D classification was used to further remove suboptimal segments, as well as to separate segments with different turn conformations; and 3D autorefinement, Bayesian polishing and contrast transfer function refinement were then repeated. The final reconstructions were sharpened using the standard postprocessing procedures in RELION-4.0, and overall resolutions were estimated from Fourier shell correlations of 0.143 between the two independently refined half-maps, using phase randomization to correct for convolution effects of a generous, soft-edged solvent mask[61]. Local resolution estimates were obtained using the same phase-randomization procedure but with a soft spherical mask that was moved over the entire map. Helical symmetry was imposed using the RELION Helix Toolbox. Further details are given in Extended Data Table 2.

### Atomic model building and refinement
The atomic models were built de novo and refined in real-space in COOT[62] using the best-resolved maps. Rebuilding using molecular dynamics was carried out in ISOLDE[63]. The models were refined in Fourier space using REFMAC5 (ref. 64), with appropriate symmetry constraints defined using Servalcat[65]. To confirm the absence of overfitting, the model was shaken, refined in Fourier space against the first half-map using REFMAC5 and compared with the second half-map. Geometry was validated using MolProbity[66]. ChimeraX[67] was used for molecular graphics and analyses. Model statistics are given in Extended Data Table 2.

### Reporting summary
Further information on research design is available in the Nature Portfolio Reporting Summary linked to this article.

### Data availability
Whole-exome data have been deposited in the National Institute on Ageing Alzheimer's Disease Data Storage Site under accession code NG00107. Mass spectrometry data have been deposited in the Proteomics Identifications database under accession numbers PXD040102 and PXD043731. Cryo-EM datasets have been deposited in the Electron Microscopy Public Image Archive under accession codes EMPIAR-11438 for individual 1, EMPIAR-11428 for individual 2 and EMPIAR-11429 for individual 3. Cryo-EM maps have been deposited in the Electron Microscopy Data Bank under accession codes EMD-16628 for individual 1, variant 1; EMD-16642 for individual 1, variant 2; EMD-16643 for individual 1, variant 3; EMD-16677 for individual 2, variant 1; EMD-16681 for individual 2, variant 2; and EMD-16682 for individual 3, variant 1. Atomic models have been deposited in the Protein Data Bank under accession codes 8CG3 for variant 1, 8CGG for variant 2 and 8CGH for variant 3. The atomic model of TDP-43 filaments from individuals with ALS and type B FTLD-TDP is available at the Protein Data Bank under accession code 7PY2.

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

**Acknowledgements** We thank the individuals and their families for donating brain tissue; the Manchester Brain Bank, which is part of the Brains for Dementia Research Initiative, jointly funded by the Alzheimer's Society and Alzheimer's Research UK, for supplying tissue from individuals 1 and 3; J. H. Grafman for supplying tissue from individual 4; the National Centralized Repository for Alzheimer's Disease and Related Dementias, which receives funding from the National Institute on Aging, for supplying tissue from individual 5; the Center for Medical Genomics of Indiana University School of Medicine for next-generation DNA sequencing; M. H. Jacobsen for help with neuropathological examinations; R. Otani for help with immunolabelling; M. Tahira for help with mass spectrometry; K. Yamashita and G. Murshudov for help with Servalcat and model refinements in REFMAC5; staff at the MRC Laboratory of Molecular Biology Electron Microscopy Facility for access to and support with cryo-EM; staff at the MRC Laboratory of Molecular Biology Scientific Computing Facility for access to and support with computing; staff at the MRC Laboratory of Molecular Biology Mass Spectrometry Facility for access to and support with mass spectrometry; and T.S. Behr, A. Bertolotti, S.W. Davies, M. Goedert, S.H.W. Scheres and S. Tetter for discussions. This work was supported by the Medical Research Council, as part of UK Research and Innovation (MC_UP_1201/25 to B.R.-F.); an Alzheimer's Research UK Rising Star Award (ARUK-RS2019-001 to B.R.-F.); the US National Institutes of Health (P30-AG010133, U01-NS110437 and RF1-AG071177 to R.V. and B.G); the Japan Agency for Medical Research and Development (JP20dm0207072 to M.H.); the Japan Science and Technology Agency Core Research for Evolutional Science and Technology (JPMJCR18H3 to M.H.); a Cambridge Commonwealth, European & International Trust Postgraduate Scholarship to R.C.; and a Leverhulme Early Career Fellowship (ECF-2022-610 to D.A.). For the purpose of open access, the MRC Laboratory of Molecular Biology has applied a CC BY public copyright licence to any author-accepted manuscript version arising.

**Author contributions** K.L.N., B.G. and M.H. identified individuals. A.C.R. and B.G. performed neuropathological examinations. A.C.R., B.G. and M.H. performed immunohistochemistry. H.J.G. and R.V. performed genetic analyses. D.A., R.C. and M.H. extracted TDP-43 filaments. D.A., F.K. and M.H. performed biochemical analyses. R.C., S.Y.P.-C. and F.K. collected and analysed mass spectrometry data. D.A. collected cryo-EM data. D.A., A.G.M. and B.R.-F. analysed cryo-EM data. B.R.-F. supervised the study. All authors contributed to the writing of the manuscript. B.G., M.H. and B.R.-F. are the senior authors.

**Competing interests** The authors declare no competing interests.

**Additional information**
**Correspondence and requests for materials** should be addressed to Benjamin Ryskeldi-Falcon.

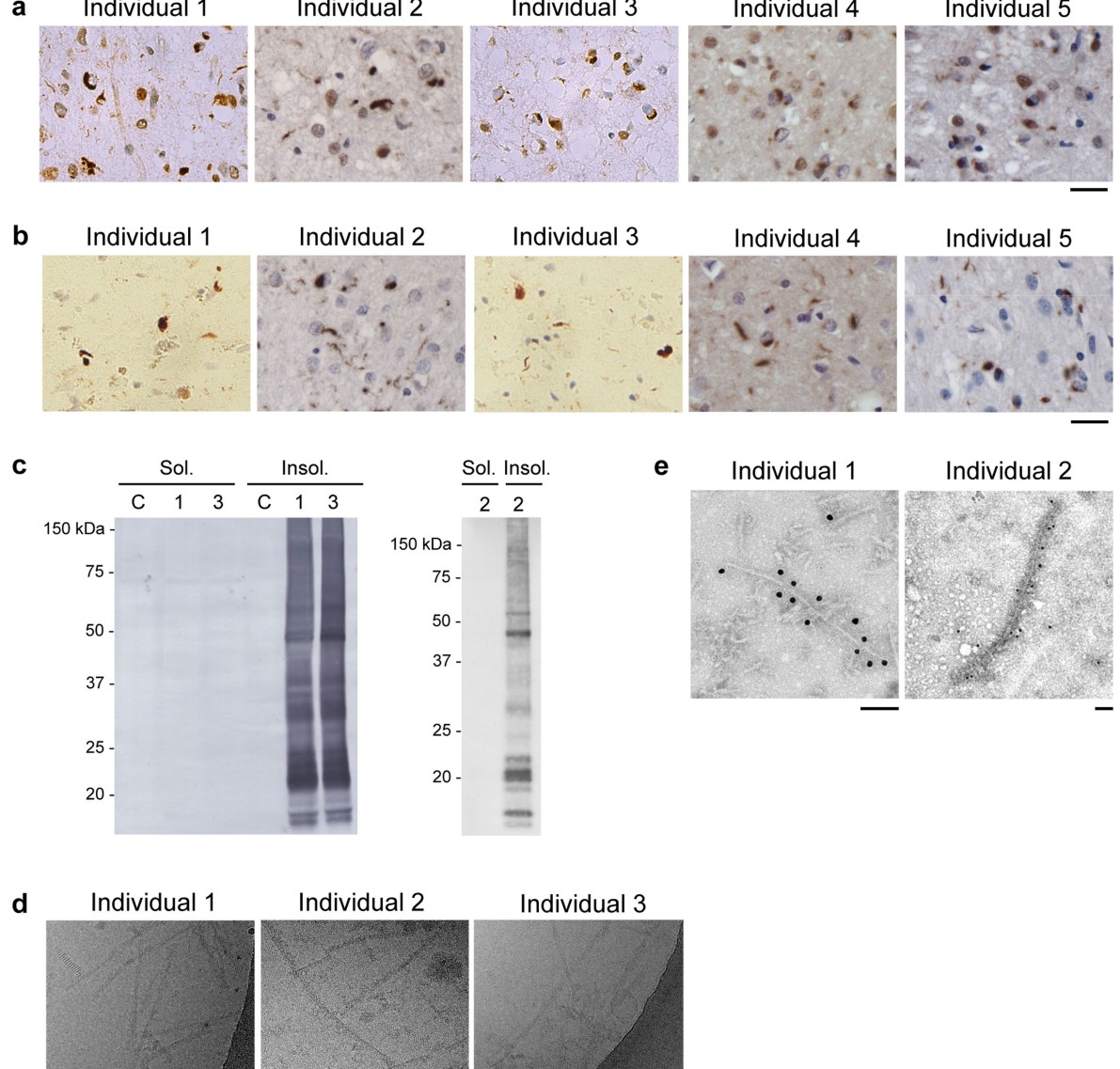

**Extended Data Fig. 1 | TDP-43 pathology of type A FTLD-TDP. a,b,** Immunohisto-chemistry for TDP-43 (brown) in the prefrontal cortex of individuals 1–5 with antibodies against TDP-43 N-terminus **(a)** and TDP-43 phosphorylated at S409 and S410 **(b)**. Sections were counterstained with hematoxylin (blue). Scale bars, 25 µm. **c,** Immunoblots of the sarkosyl-soluble and sarkosyl-insoluble fractions from the prefrontal cortex of individuals 1–3 and a control case without TDP-43 pathology (C) with an antibody against TDP-43 phosphorylated at S409 and S410. For gel source data, see Supplementary Fig. 1. **d,** Cryo-EM images of TDP-43 filaments from the prefrontal cortex of individuals 1–3. Scale bar, 50 nm. **e,** Immuno-gold negative-stain EM images of TDP-43 filaments from the prefrontal cortex of individuals 1 and 2 with an antibody against TDP-43 phosphorylated at S409 and S410. Scale bars, 100 nm. For **a–e,** similar results were obtained in at least three independent experiments.

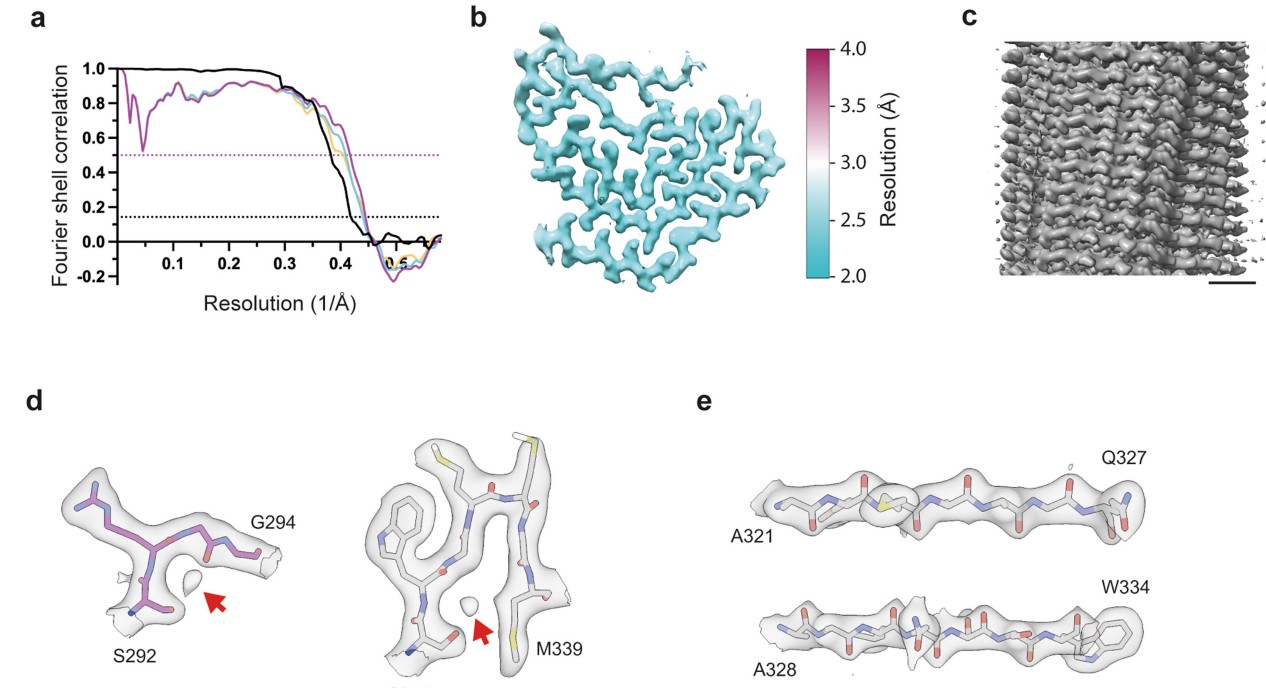

**a** Fourier shell correlation vs Resolution (1/Å)

**b** Resolution (Å) 4.0 – 2.0

**c**

**d** G294, S292, S333, M339

**e** Q327, A321, W334, A328

**Extended Data Fig. 2 | Cryo-EM density map and atomic model. a**, Fourier shell correlation (FSC) curves for the two independently-refined cryo-EM half-maps of TDP-43 filaments from Type A FTLD-TDP (black line); for the refined atomic model against the cryo-EM density map (magenta); for the atomic model shaken and refined using the first half-map against the first half-map (cyan); and for the same atomic model against the second half-map (yellow). FSC thresholds of 0.143 (black dashed line) and 0.5 (magenta dashed line) are shown. **b**, Local resolution estimate for the cryo-EM density map. **c**, Cryo-EM density map viewed along the helical axis. Scale bar, 10 Å. **d,e**, Views of the cryo-EM density map and atomic model showing representative densities for ordered solvent (red arrows) (**d**) and main chain oxygen atoms in β-strands, which reveal the chirality of the map (**e**).

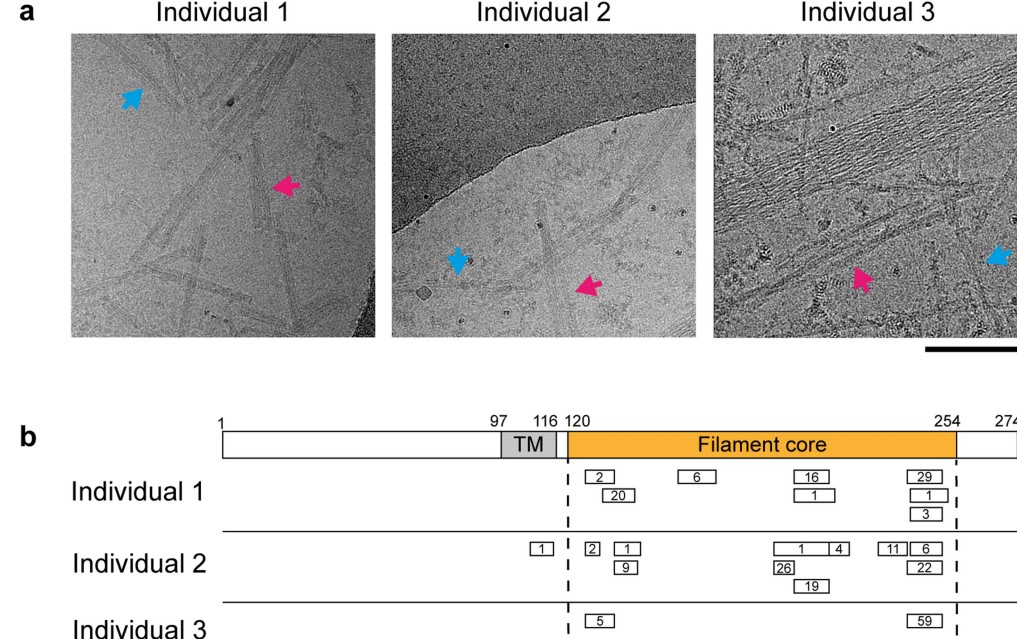

**Extended Data Fig. 3 | TMEM106B filaments from type A FTLD-TDP.**
**a**, Cryo-EM images of TMEM106B filaments from the prefrontal cortex of individuals 1–3. Arrows indicate examples of single (cyan) and double (magenta) TMEM106B filaments. Scale bars, 100 nm. Similar results were obtained in at least three independent experiments. **b**, Mass spectrometry TMEM106B peptide coverage from the prefrontal cortex of individuals 1-3. The peptides map to the region that forms the ordered core of TMEM106B filaments, with the exception of one peptide from individual 2, which maps to the transmembrane helix (TM) of TMEM106B. The counts are given for each peptide. Peptide sequences are shown in Supplementary Data Table 1.

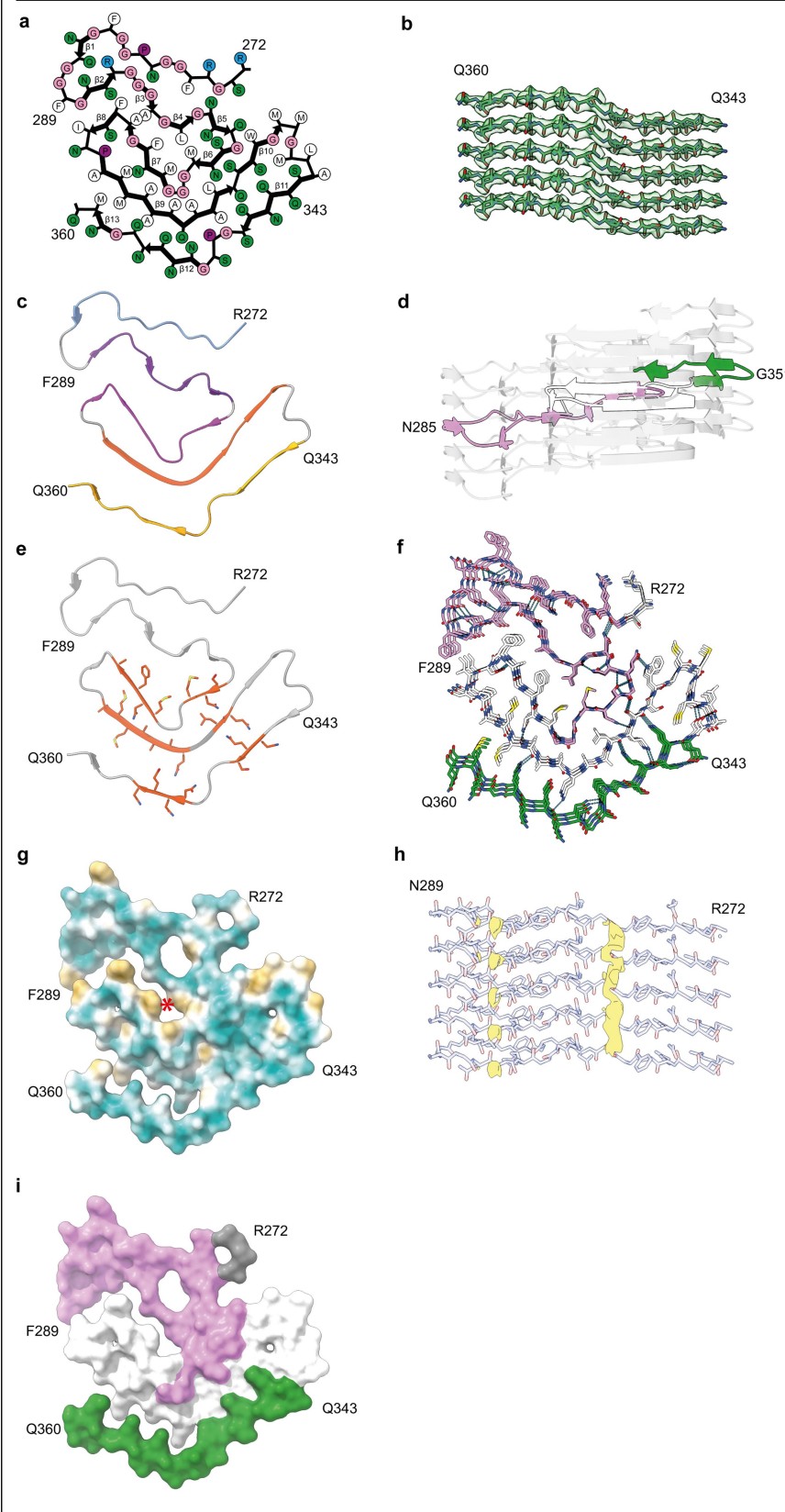

**Extended Data Fig. 4 | The TDP-43 filament fold of type A FTLD-TDP. a**, Schematic representation of the type A FTLD-TDP fold. **b**, Cryo-EM map and atomic model, shown for the Q/N-rich region of five TDP-43 molecules in line with the helical axis. **c**–**e**, Secondary structure of the type A FTLD-TDP fold, shown for a single TDP-43 molecule perpendicular to the helical axis with the five layers highlighted in different colours **(c)**; shown for five TDP-43 molecules in line with the helical axis **(d)**; and shown for a single TDP-43 molecule perpendicular to the helical axis with regions of zipper packing highlighted in orange **(e)**. **f**, Atomic model of the fold depicting intramolecular hydrogen bonds (cyan dashed lines), shown for three TDP-43 molecules perpendicular to the helical axis. **g**, Hydrophobicity of the fold from most hydrophilic (cyan) to most hydrophobic (yellow), shown for a single TDP-43 molecule perpendicular to the helical axis. The red asterisk indicates the hydrophobic cluster formed at the interface of the second and third layers. **h**, View of the atomic model and non-proteinaceous densities, shown for five TDP-43 molecules in line with the helical axis. **i**, Surface representation of the fold, shown for a single TDP-43 molecule perpendicular to the helical axis. For **b,d,f**, the glycine-rich (G274–G310, magenta), hydrophobic (M311–S342, white) and Q/N-rich (Q343–Q360, green) regions are highlighted.

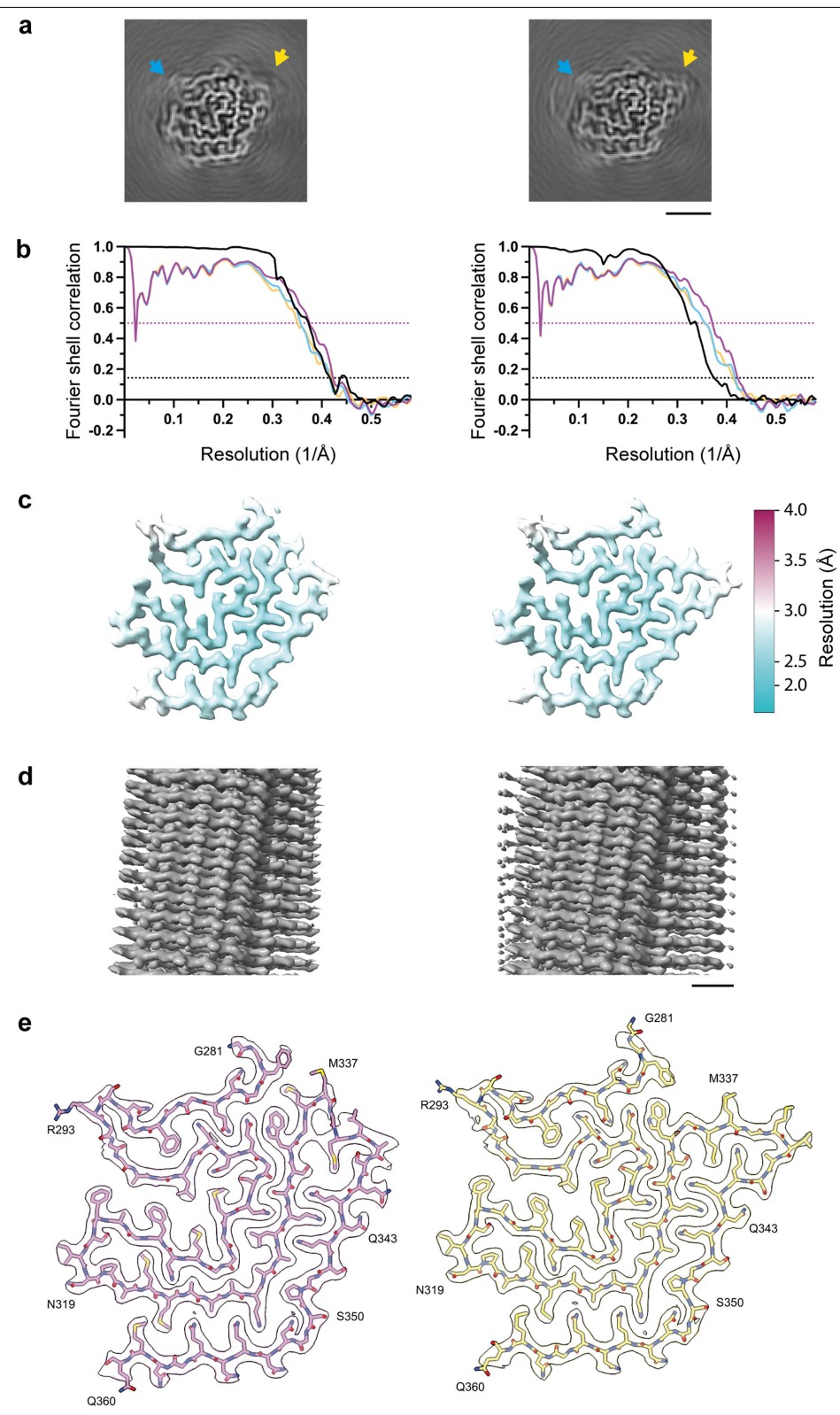

**Extended Data Fig. 5 | Alternative local conformations of the type A FTLD-TDP filament fold. a**, Cryo-EM maps of TDP-43 filaments from type A FTLD-TDP with different local conformations of the N-terminal region (cyan arrows) and of the turn connecting the fourth layer to the fifth (yellow arrows), shown as central slices perpendicular to the helical axis. Scale bars, 25 Å. **b**, Fourier shell correlation (FSC) curves for the two independently-refined cryo-EM half-maps (black lines); for the refined atomic model against the cryo-EM density map (magenta); for the atomic model shaken and refined using the first half-map against the first half-map (cyan); and for the same atomic model against the second half-map (yellow). FSC thresholds of 0.143 (black dashed line) and 0.5 (magenta dashed line) are shown. **c**, Local resolution estimates for the cryo-EM density maps. **d**, Cryo-EM density maps viewed along the helical axis. Scale bar, 10 Å. **e**, Cryo-EM density maps and atomic models, shown for a single TDP-43 molecule perpendicular to the helical axis.

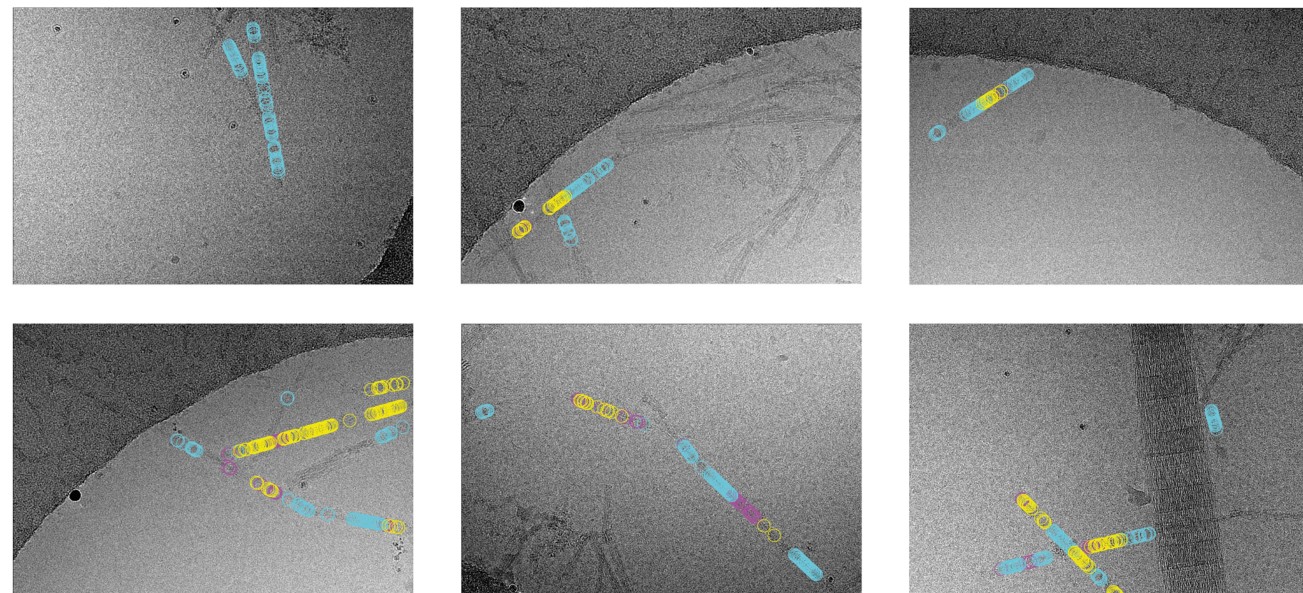

**Extended Data Fig. 6 | Structural variation within individual TDP-43 filaments from type A FTLD-TDP.** Example micrographs of TDP-43 filaments from type A FTLD-TDP individual 1 showing the positions of filament segments contributing to cryo-EM maps with different local conformations, which can occur within individual filaments. Filament segments with the main local conformations are shown in cyan; with the alternative local conformation of the N-terminal region in magenta; and with the alternative local conformation of the N-terminal region and the turn connecting the fourth layer to the fifth in yellow. Scale bar, 100 nm.

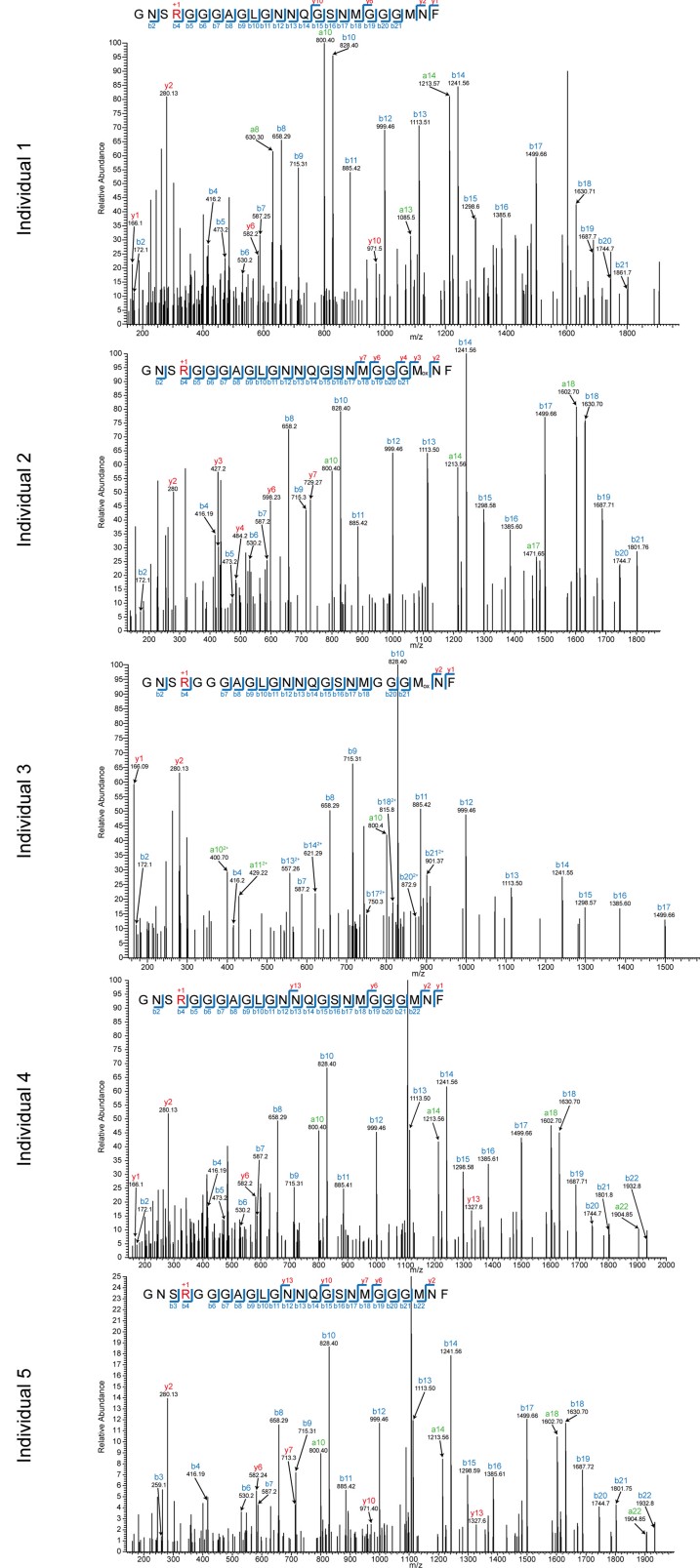

**Extended Data Fig. 7 | Citrullination of R293 in TDP-43 filaments from type A FTLD-TDP.** Mass spectra of TDP-43 peptides containing citrullinated R293 from TDP-43 filaments isolated from the prefrontal cortex of individuals 1–5.

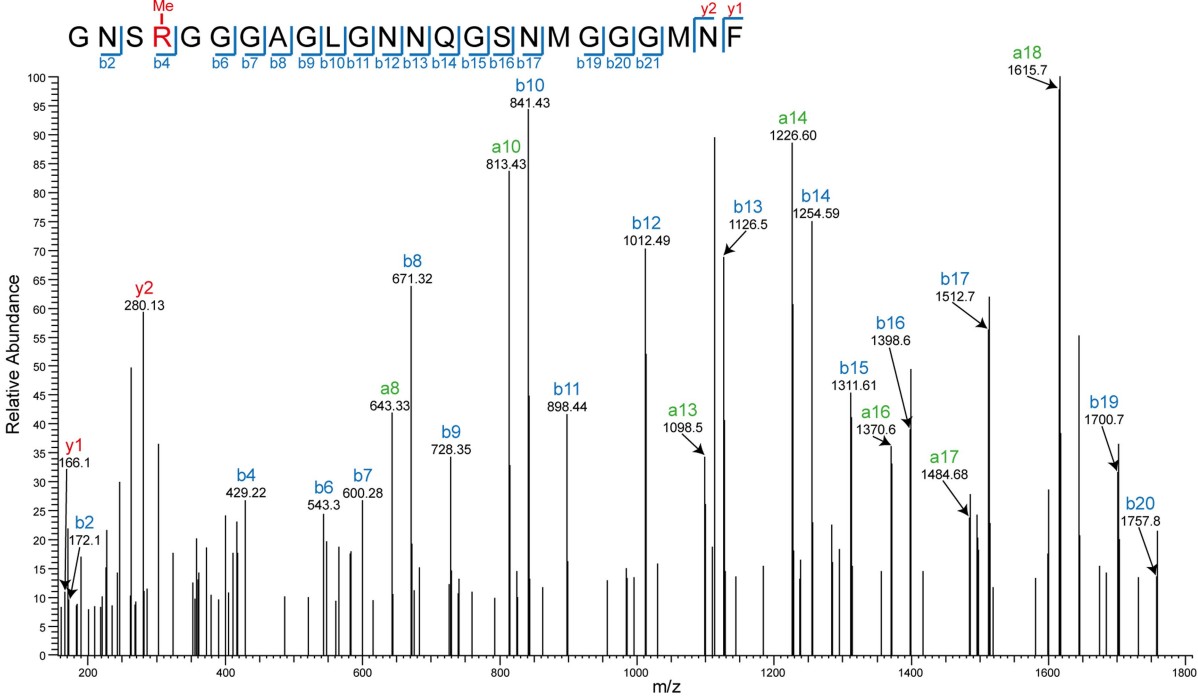

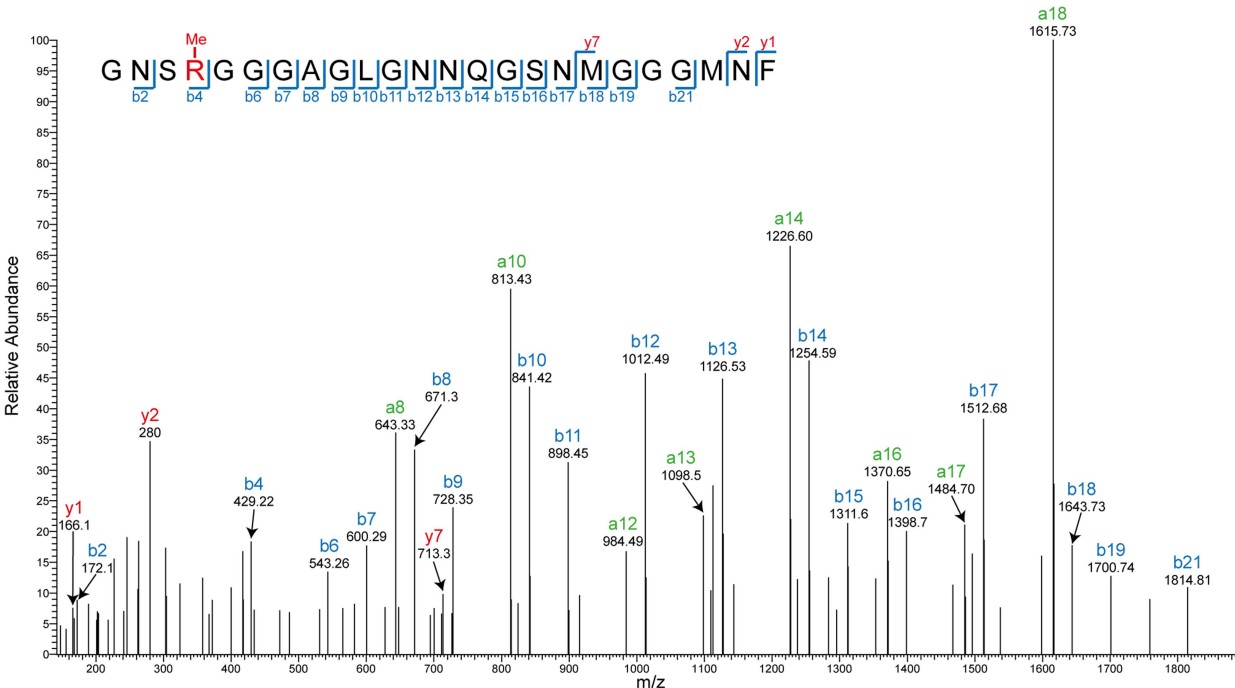

**Extended Data Fig. 8 | Mono-methylation of R293 in TDP-43 filaments from type A FTLD-TDP.** Mass spectra of TDP-43 peptides containing mono-methylated R293 from TDP-43 filaments isolated from the prefrontal cortex of individuals 1 and 2. Such peptides were not detected for individuals 3–5, possibly because fewer total peptides containing R293 of TDP-43 were detected for these individuals compared to individuals 1 and 2.

**Extended Data Table 1 | Clinicopathological details**

|  | Individual 1 | Individual 2 | Individual 3 | Individual 4 | Individual 5 |
|---|---|---|---|---|---|
| **Clinicopathological evaluation** | | | | | |
| Male/ female | M | F | M | F | F |
| Age (y) | 72 | 50 | 71 | 59 | 75 |
| Disease duration (y) | 10 | 5 | 7 | 4 | 4 |
| Clinical diagnosis | FTD | FTD | FTD | FTD | FTD |
| Neuropathological diagnosis | Type A FTLD-TDP | Type A FTLD-TDP | Type A FTLD-TDP | Type A FTLD-TDP | Type A FTLD-TDP |
| Tau pathology (Braak) | ND | ND | ND | I | 0 |
| Amyloid-β pathology (Thal) | 0 | ND | 0 | 3 | 1 |
| α-synuclein pathology | ND | ND | ND | ND | ND |
| C9orf repeats | WT | WT | WT | WT | WT |
| *GRN* | c.1354del (p.V452WfsX38) | c.1477C>T (p.R493X) | WT | c.709-2A>G (splice acceptor) | c.1420_1421del (p.C474fs) |

M, male; F, female; FTD, frontotemporal dementia; FTLD-TDP, frontotemporal lobar degeneration with TDP-43 pathology; ND, not detected.

**Extended Data Table 2 | Cryo-EM data collection, refinement and validation statistics**

| | Individual 1 (EMPIAR-11438) (EMDB-16628, EMD-16642 and EMD-16643) (PDB 8CG3, 8CGG and 8CGH) | | | Individual 2 (EMPIAR-11428) (EMD-16677 and EMD-16681) | | | Individual 3 (EMPIAR-11429) (EMD-16682) | | |
|---|---|---|---|---|---|---|---|---|---|
| **Data collection and processing** | | | | | | | | | |
| Voltage (kV) | 300 | | | 300 | | | 300 | | |
| Electron source | XFEG | | | XFEG | | | XFEG | | |
| Detector | K3 | | | K3 | | | K3 | | |
| Electron exposure (e–/Å$^2$) | 35.896 | | | 33.681 | | | 29.9 | | |
| Defocus range (μm) | -2.2 to -1.0 | | | -2.2 to -1.0 | | | -2.2 to -1.0 | | |
| Pixel size (Å) | 0.86 | | | 0.86 | | | 0.86 | | |
| Symmetry imposed | C1 | | | C1 | | | C1 | | |
| Initial particle images (no.) | 996,193 | | | 210,906 | | | 238,532 | | |
| | **V1** | **V2** | **V3** | **V1** | **V2** | **V3** | **V1** | **V2** | **V3** |
| Final particle images (no.) | 71,963 | 19,957 | 16,260 | 100,937 | 4,987 | - | 28,943 | - | - |
| Helical twist (°) | 1.175 | 1.229 | 1.248 | 1.157 | 1.232 | - | 1.151 | - | - |
| Helical rise (Å) | 4.996 | 4.996 | 4.98 | 4.99 | 4.996 | - | 4.974 | - | - |
| Map resolution (Å) | 2.4 | 2.5 | 2.7 | 2.5 | 2.9 | - | 2.9 | - | - |
| FSC threshold | 0.143 | 0.143 | 0.143 | 0.143 | 0.143 | | 0.143 | | |
| Map resolution range (Å) | 2.3–4.1 | 2.5–6.7 | 2.5–5.7 | 2.5–4.6 | 2.8–6.6 | - | 2.8–5.8 | - | - |
| **Refinement** | | | | | | | | | |
| Initial model used (PDB code) | - | - | - | - | - | - | - | - | - |
| Model resolution (Å) | 2.4 | 2.7 | 2.7 | - | - | - | - | - | - |
| FSC threshold | 0.5 | 0.5 | 0.5 | | | | | | |
| Map sharpening $B$ factor (Å$^2$) | -36.1 | -36.23 | -39.85 | - | - | - | - | - | - |
| Model composition | | | | - | - | - | - | - | - |
| Non-hydrogen atoms | 609 | 543 | 543 | | | | | | |
| Protein residues | 89 | 80 | 80 | | | | | | |
| Ligands | - | - | - | | | | | | |
| $B$ factors (Å$^2$) | | | | - | - | - | - | - | - |
| Protein | 53.57 | 51.31 | 46.49 | | | | | | |
| Ligand | - | - | - | | | | | | |
| R.m.s. deviations | | | | - | - | - | - | - | - |
| Bond lengths (Å) | 0.008 | 0.007 | 0.007 | | | | | | |
| Bond angles (°) | 1.71 | 1.64 | 1.47 | | | | | | |
| Validation | | | | - | - | - | - | - | - |
| MolProbity score | 0.99 | 1.84 | 1.56 | | | | | | |
| Clashscore | 1.73 | 4.85 | 2.01 | | | | | | |
| Poor rotamers (%) | 0 | 0 | 0 | | | | | | |
| Ramachandran plot | | | | - | - | - | - | - | - |
| Favored (%) | 97.7 | 88.46 | 92.31 | | | | | | |
| Allowed (%) | 100 | 100 | 100 | | | | | | |
| Outliers (%) | 0 | 0 | 0 | | | | | | |

V1, map of TDP-43 filaments with the main conformation; V2, map of TDP-43 filaments with the alternative conformation of the N-terminal region; V3, map of TDP-43 filaments with alternative conformations of the N-terminal region and of the turn connecting the fourth layer to the fifth.

# Reporting Summary

## Statistics

For all statistical analyses, confirm that the following items are present in the figure legend, table legend, main text, or Methods section.

| n/a | Confirmed | |
|---|---|---|
| ☐ | ☒ | The exact sample size (*n*) for each experimental group/condition, given as a discrete number and unit of measurement |
| ☐ | ☒ | A statement on whether measurements were taken from distinct samples or whether the same sample was measured repeatedly |
| ☒ | ☐ | The statistical test(s) used AND whether they are one- or two-sided<br>*Only common tests should be described solely by name; describe more complex techniques in the Methods section.* |
| ☒ | ☐ | A description of all covariates tested |
| ☒ | ☐ | A description of any assumptions or corrections, such as tests of normality and adjustment for multiple comparisons |
| ☐ | ☒ | A full description of the statistical parameters including central tendency (e.g. means) or other basic estimates (e.g. regression coefficient) AND variation (e.g. standard deviation) or associated estimates of uncertainty (e.g. confidence intervals) |
| ☒ | ☐ | For null hypothesis testing, the test statistic (e.g. *F*, *t*, *r*) with confidence intervals, effect sizes, degrees of freedom and *P* value noted<br>*Give P values as exact values whenever suitable.* |
| ☒ | ☐ | For Bayesian analysis, information on the choice of priors and Markov chain Monte Carlo settings |
| ☒ | ☐ | For hierarchical and complex designs, identification of the appropriate level for tests and full reporting of outcomes |
| ☒ | ☐ | Estimates of effect sizes (e.g. Cohen's *d*, Pearson's *r*), indicating how they were calculated |

*Our web collection on statistics for biologists contains articles on many of the points above.*

## Software and code

Policy information about availability of computer code

| Data collection | EPU 2.14 |
|---|---|
| Data analysis | Mascot 2.4, Scaffold 4.0, RELION 4.0, CTFFIND 4.1, COOT 0.9.8.2, ISOLDE 1.5, REFMAC 5.8.0387, Servalcat 0.3.0, ChimeraX 1.5, MolProbity 4.5.2. |

For manuscripts utilizing custom algorithms or software that are central to the research but not yet described in published literature, software must be made available to editors and reviewers. We strongly encourage code deposition in a community repository (e.g. GitHub). See the Nature Portfolio guidelines for submitting code & software for further information.

## Data

Policy information about availability of data

All manuscripts must include a data availability statement. This statement should provide the following information, where applicable:
- Accession codes, unique identifiers, or web links for publicly available datasets
- A description of any restrictions on data availability
- For clinical datasets or third party data, please ensure that the statement adheres to our policy

Whole-exome data have been deposited in the National Institute on Ageing Alzheimer's Disease Data Storage Site (NIAGADS), under accession code NG00107. Mass spectrometry data have been deposited to the Proteomics Identifications (PRIDE) database under the accession numbers PXD040102 and PDX043731. Cryo-EM datasets have been deposited to the Electron Microscopy Public Image Archive (EMPIAR) under accession codes EMPIAR-11438 for individual 1, EMPIAR-11428 for

# Human research participants

Policy information about studies involving human research participants and Sex and Gender in Research.

| | |
|---|---|
| Reporting on sex and gender | 3 females and 2 males. |
| Population characteristics | See Extended Data Table 1. Between 50 and 75 years-of-age. Disease associated mutations in GRN. Wild-type C9orf. Clinical diagnoses of FTD. Neuropathological diagnoses of type A FTLD-TDP. |
| Recruitment | Selected based on availability and neuropathological examination. |
| Ethics oversight | Human tissue samples were from the Brain Library of the Dementia Laboratory at Indiana University School of Medicine (individuals 2, 4, and 5) and the Manchester Brain Bank (individuals 1 and 3). Their use in this study was approved by the ethical review processes at each institution. Informed consent was obtained from the patients' next of kin. |

Note that full information on the approval of the study protocol must also be provided in the manuscript.

# Field-specific reporting

Please select the one below that is the best fit for your research. If you are not sure, read the appropriate sections before making your selection.

☒ Life sciences ☐ Behavioural & social sciences ☐ Ecological, evolutionary & environmental sciences

For a reference copy of the document with all sections, see nature.com/documents/nr-reporting-summary-flat.pdf

# Life sciences study design

All studies must disclose on these points even when the disclosure is negative.

| | |
|---|---|
| Sample size | Prefrontal cortex from 3 individuals with type A FTLD-TDP. Samples were chosen based on availability and neuropathological examination. |
| Data exclusions | Pre-established common image classification procedures (Scheres 2012. J. Struc. Biol. 180, 519-530) were employed to select particle images with the highest resolution content in the cryo-EM reconstruction process. Details of the number of selected images are given in Extended Data Table 2. |
| Replication | All attempts at replication were successful. At least three independent biological repeats per experiment where representative data are shown, as described in the main text. |
| Randomization | Randomisation was not performed. As the samples were limited by brain availability, randomisation would not have reduced any bias in this study. |
| Blinding | The investigators were not blinded to allocation during experiments and outcome assessment. The perceived risk of detection/performance bias was deemed negligible. |

# Reporting for specific materials, systems and methods

We require information from authors about some types of materials, experimental systems and methods used in many studies. Here, indicate whether each material, system or method listed is relevant to your study. If you are not sure if a list item applies to your research, read the appropriate section before selecting a response.

## Materials & experimental systems

| n/a | Involved in the study |
|-----|----------------------|
| ☐ | ☒ Antibodies |
| ☒ | ☐ Eukaryotic cell lines |
| ☒ | ☐ Palaeontology and archaeology |
| ☒ | ☐ Animals and other organisms |
| ☒ | ☐ Clinical data |
| ☒ | ☐ Dual use research of concern |

## Methods

| n/a | Involved in the study |
|-----|----------------------|
| ☒ | ☐ ChIP-seq |
| ☒ | ☐ Flow cytometry |
| ☒ | ☐ MRI-based neuroimaging |

## Antibodies

| | |
|---|---|
| Antibodies used | The primary antibodies used were anti-phospho S409 and S410 TDP-43 (Cosmo Bio USA, CAC-TIP-PTD-M01, clone 11-9), anti-N-terminus TDP-43 (Abcam, ab57105) and anti-N-terminus TDP-43 (Proteintech, 10782-2-AP). |
| Validation | Validation of anti-phospho S409 and S410 TDP-43 against human phospho S409 and S410 TDP-43 for histology and immunoblotting is presented in the manufacturer's datasheet (Cosmo Bio USA) and in (Inukai et al. 2008. FEBS Lett. 582, 2899-2904). Validation of anti-N-terminus TDP-43 against human TDP-43 for histology and immunoblotting is presented in the manufacturers' datasheets (Abcam and Proteintech). |

