## [Peer Review File · Nature]

Manuscript Title: TDP-43 forms amyloid filaments with a distinct fold in type A FTLD-TDP

Reviewer Comments & Author Rebuttals

Reviewer Reports on the Initial Version:

Referees' comments:

Referee #1 (Remarks to the Author):

The RNA-binding protein TDP-43 forms pathological aggregates in the brains of patients with amyotrophic lateral sclerosis and frontotemporal dementia. Since its discovery in 2006 and the initial observations that unlike most of the other proteins that accumulate in other neurodegenerative diseases, most TDP-43 aggregates do not stain with amyloidophilic dyes, a debate in the field has been whether TDP-43 aggregates in disease adopt amyloid fibrillar structures or not. This debate has become even stronger last year, when a number of studies identified amyloid fibrils of another protein, called TMEM106B, in tissues of patients with ALS and FTD, but also those of patients with other neurodegenerative diseases, as well as old subjects without evidence of a neurodegenerative condition. In previous work, the team has used ALS patient tissues, as well as tissues from FTLD-TDP with the subtype B to resolve the structures of TDP-43 aggregates and they reported that they adopt a fibrillar structure of double-spiral-shaped fold with no obvious structural distinction between ALS and FTD type B. In the current study Arseni et al resolve TDP-43 filament structures from three patient brain samples with type-A FTLD-TDP using cryo-EM. They show that type A TDP-43 fibrils have a distinct structure from the ALS and FTLD-TDP fibrils. Using mass spectrometry, the team identifies two new post-translational modifications of assembled TDP-43, citrullination, and mono-methylation of R293, which they propose to facilitate the formation of the fibrils.

This is an excellent piece of work with high impact on the field since these data unambiguously show that 1. TDP-43 accumulations in patients adopt an amyloid fibrillary structure and 2. TDP-43 fibrils from different disease subtypes have distinct molecular structures. This is in line with data from other neurodegenerative diseases and suggests that the different fibrillar structures trigger different toxic pathways in the patients with different disease subtypes, in agreement with the notion of "TDP-43 strains" that was previously proposed. For this reason, I enthusiastically recommend the work for publication in Nature, after revision of the current manuscript to address the following points of criticism.

Main points of criticism:

1. An important point that the authors should clarify is the discrepancy of the current study and that of Jiang et al., 2022 (<https://doi.org/10.1038/s41586-022-04670-9>). The question is: what is (if any) the role of TMEM106B in disease and the potential interplay between TMEM106B and TDP-43 fibrils? Here are my specific recommendations on this important point. In lines 99-101, the authors state: "The presence of both TDP-43 filaments and TMEM106B filaments in type A FTLD-TDP, as well as in ALS with type B FTLD-TDP9, is at odds with a recent report of amyloid filaments in FTLD-TDP being composed of TMEM106B, but not TDP-43 (ref 19). Differences in the brain extraction protocol used in that study, including a lower centrifugation speed used to pellet the material examined, may

have been the reason TDP-43 filaments were not observed.” However, if this was the case, then they should not see non-fibrillar TDP-43 aggregates as it is expected that non-fibrillar (protofibrillar or pathological oligomers) aggregates would settle or sediment at higher centrifugation speeds. Yet, Jiang et al. report that they do see these in their samples (See Fig. 1c of Jiang et al. paper). Therefore, it would be important for the field to clarify the differences and similarities in the preparation or, potentially sample heterogeneity that may account for the differences. Some specific questions/suggestions on this point are the following:

- a. Is there any additional reasoning the authors could provide to support why fibrils are not detected by Jiang et al., 2022, but are in the current study?
- b. Authors need to clarify that their extraction procedure does not induce any TDP-43 fibril formation.
- c. Do the authors observe any non-fibrillar soluble TDP-43 species (pathological oligomeric TDP-43 species) in their preparations, similar to those reported by Jiang et al? To this end, authors should show negative stain TEM images as those are missing from the figures.
- d. The authors should show both the supernatant and pellet fractions in the immunoblots to confirm or rule out the possibility of the absence of non-fibrillar TDP-43 aggregates and also to solidify their results that type-A FTLD patients have abundant fibrillar TDP-43 species.
- e. Authors should perform immunoelectron microscopy (similar to what they have done in <https://doi.org/10.1007/s00401-012-1055-8>) with TDP-specific to show that fibrils can be detected in the cells of the brain sections they used for cryo-EM. This way, they can strengthen their claims of detecting TDP-43 fibrils. The same analysis using TMEM106B-specific antibodies would be extremely interesting to explore whether TMEM106B and TDP-43 fibrils may co-aggregate.

2. In (line 180, 181, page 7), the authors state that “Analysis of filaments from the brains of three individuals with type A FTLD-TDP by mass spectrometry established the presence of TDP-43 molecules that were citrullinated at R293”. Can the authors clarify whether this PTM is observed exclusively in filaments or is it also possible to detect these in the non-fibrillar TDP-43 aggregates?

3. In lines 86-87, the authors state that “The filaments comprised a single amyloid protofilament of stacked TDP-43 molecules.” How does this explain the extra density viewed beyond R272? Is it possible that they are also appearing due to parallelly aligned protofilament? I note that data showing that the RRM fragment fits this density better than an adjacent fibril will clarify the point.

4. In lines 94-96, the authors state that: “The presence of TMEM106B was confirmed using mass spectrometry, which identified peptides mapping to the C-terminal region that forms the ordered core of the filaments (Extended Data Fig. 3b)19–21.” The authors should clarify whether the mass spectrometry analysis of the TMEM106B fibrils was performed from tissues of all 3 individuals.

5. In line 105, the authors claim that “There are no clear relationships between TMEM106B filament folds and diseases.” While I think that the authors refer to other studies suggesting that TMEM106B fibrils occur in an age-dependent manner in the brains of individuals that did not suffer from neurodegenerative diseases, one cannot exclude the possibility that TMEM106B and TDP-43 interact in a pathologic manner in patient brains, perhaps even co-aggregating. This possibility should be discussed or experimentally tested by the authors.

6. In lines 111-112 the authors state: “The fold consists of five connected layers formed by R272–Q360 in the LCD, with flanking regions forming the fuzzy coat.” Can the electron densities beyond R272 be fit to the residues of LCD which could contribute to the fuzzy coat?

7. In lines 133-134, the authors claim that “This [less ordered protein] density could accommodate approximately 18 residues (S255–E271), which would include b5 of the second TDP-43 RRM”. It would be interesting to see modeling simulation data for this prediction or at least data predicting the propensity of this region to fold into an amyloid-like fold.

8. In lines 134-136, the authors state that “The incorporation of CTFs that terminate between S255–E271 into the filaments may explain why this density was less ordered²⁷. Their incorporation would lead to the exposure of the hydrophobic patch.” Shouldn’t the incorporation of the residues from S255-S271 lead to the burial, rather than exposure of the hydrophobic patch? Maybe some structure prediction data of these residues aligning next to the patch could clarify the point.

9. In lines 139-141, the authors conclude that “An additional peptide-like density is located on the outside surface of the 139 fifth layer, adjacent to G351–N355 (Fig. 1a,d). Its disconnected nature precluded sequence assignment. It may originate from the N- or C-terminal flanking regions, or from a separate interacting protein.” It would be very interesting to check if residues from TMEM106B could fit here.

10. In line 215 the authors state that “Residues A321–Q331 in the hydrophobic region form the nexus of both filament folds”. TDP-43 linked mutations are also found on residues A321 and Q331. The authors should comment on the possible effect of these mutations on the structure of the core of the filaments.

Minor points:

1. Authors should comment on the contribution of different cell types or brain regions in driving different amyloid filament folds.

2. Mass spec data belongs to individuals 2, 4, 5. Since the structures obtained are mainly from individuals 1,2,3 it would be helpful to add mass spectrometry data from individuals 1 and 3.

3. On line 49 the authors state referring to: “They bind the dye thioflavin-T poorly¹⁸”. This should be thioflavin-S, not thioflavin-T.

4. The authors should indicate which part of the frontal cortex (is it the motor cortex or prefrontal cortex) was used for brain extraction.

Referee #2 (Remarks to the Author):

Nature 2023-02-02272

TDP-43 forms amyloid filaments with a distinct fold in type A FTLD-TDP

Arseni et al.

I would say at the outset that my expertise is in cryoEM and structural biology, not in FTLD or TDP-43-associated disorders.

The manuscript describes *ex vivo* structures of TDP-43 amyloid fibrils from the brains of three individuals with frontotemporal lobar degeneration with TDP-43 pathology, but who do not have accompanying ALS pathology. I have very few issues with the work as presented. This is a concise and succinct, well-written manuscript with generally clearly presented results and conclusions. The mosaicity of post-translationally modified subunits along individual fibrils is fascinating for example. I do have some specific queries and suggestions that follow, but none of them should preclude publication of the work. However, I am not convinced that the study generates sufficient new biological or medical insight to merit publication in Nature.

The fundamental insight reported is that a protein adopts a different fold in amyloid aggregates in the brains of individuals with different pathologies. However, this now seems to me to be well established – in large part in the pages of Nature, where pathology specific *ex vivo* structures have already been reported for Tau, A β and α -synuclein for example. So, my reaction to this well executed study on TDP-43 was: yes, I assumed this might well be the case. The study therefore feels fundamentally incremental rather field-shifting.

I offer some constructive criticism that I hope will help improve the work regardless of where it is published.

1. On L111, the authors state the fibril has a right-handed twist, and cite Fig1b-d, ED Fig 2d-e. However, these figures speak to the quality of the map/model, not its handedness. I agree that the handedness of the structure should just about be discernible at $\sim 2.4\text{\AA}$, but there is no evidence of this that I can see in any figure or methods section.
2. I was interested in the non-protein densities in the structure (Fig1d) – do these span multiple amyloid layers? – note I use layer in the 4.8\AA spacing along the fibril helical axis, not the ‘five layers within the fold’ sense the authors use. It’s impossible to see this from the figures/text. Indeed the only side-view of the fibril I found was in ED data where the threshold used was quite generous – not allowing the reader to see clear layer separation as I expect there is at this resolution
3. Indeed, I found the ‘five-layer’ description unhelpful in a structural field where ‘layer’ normally refers to the 4.8\AA spacing along the fibril axis. This was perhaps more so because it’s not really used or explained in their figures – if the authors feel this terminology is useful, I would suggest it is established in Figure 1.
4. The alternate local conformations in Figure 2 are very cool – as is the MS backup to show the PTMs involved.

5.

Minor Points.

1. Personally I find it very helpful for the data availability statement to directly link which Accession code relates to which structure
2. Do the authors intend to deposit the raw micrograph movies in EMPIAR? There is no reference to this in the data accessibility statement If not why not?

Referee #3 (Remarks to the Author):

In this paper, Ryskeldi-Falcon and coworkers present cryo-EM structures of TDP-43 fibrils isolated from type A FTLD-TDP. Remarkably, these fibrils adopt a different conformation than TDP-43 fibrils formed in ALS and type B FTLD-TDP. They also find TMEM106B fibrils in type A FTLD-TDP patient extracts, which are anticipated from prior studies. Intriguingly, they find a buried arginine (R293) in the fibril core, which might be predicted to be destabilizing. However, they also find that R293 is likely citrullinated in assembled fibrils, which may explain how they form. They also find that in some TDP-43 molecules, R293 can be monomethylated, but this modification is likely incompatible with the fibril core, which is consistent with in vitro findings that monomethylation of TDP-43 at R293 can reduce TDP-43 aggregation. Overall, these studies are a tour de force and establish the formation of distinct TDP-43 fibril 'strains' in distinct neurodegenerative disorders. They set the stage for therapeutic strategies to target the distinct forms of assembled TDP-43 in distinct disorders. I recommend publication without delay.

Author Rebuttals to Initial Comments:

We thank the Referees for their constructive and insightful comments, which we feel have made a great improvement to the manuscript. Please find our point-by-point responses to the individual comments below, written in blue text.

Referee #1 (Remarks to the Author):

The RNA-binding protein TDP-43 forms pathological aggregates in the brains of patients with amyotrophic lateral sclerosis and frontotemporal dementia. Since its discovery in 2006 and the initial observations that unlike most of the other proteins that accumulate in other neurodegenerative diseases, most TDP-43 aggregates do not stain with amyloidophilic dyes, a debate in the field has been whether TDP-43 aggregates in disease adopt amyloid fibrillar structures or not. This debate has become even stronger last year, when a number of studies identified amyloid fibrils of another protein, called TMEM106B, in tissues of patients with ALS and FTD, but also those of patients with other neurodegenerative diseases, as well as old subjects without evidence of a neurodegenerative condition. In previous work, the team has used ALS patient tissues, as well as tissues from FTLN-TDP with the subtype B to resolve the structures of TDP-43 aggregates and they reported that they adopt a fibrillar structure of double-spiral-shaped fold with no obvious structural distinction between ALS and FTD type B. In the current study Arseni et al resolve TDP-43 filament structures from three patient brain samples with type-A FTLN-TDP using cryo-EM. They show that type A TDP-43 fibrils have a distinct structure from the ALS and FTLN-TDP fibrils. Using mass spectrometry, the team identifies two new post-translational modifications of assembled TDP-43, citrullination, and mono-methylation of R293, which they propose to facilitate the formation of the fibrils. This is an excellent piece of work with high impact on the field since these data unambiguously show that 1. TDP-43 accumulations in patients adopt an amyloid fibrillary structure and 2. TDP-43 fibrils from different disease subtypes have distinct molecular structures. This is in line with data from other neurodegenerative diseases and suggests that the different fibrillar structures trigger different toxic pathways in the patients with different disease subtypes, in agreement with the notion of “TDP-43 strains” that was previously proposed. For this reason, I enthusiastically recommend the work for publication in Nature, after revision of the current manuscript to address the following points of criticism.

Main points of criticism:

1. An important point that the authors should clarify is the discrepancy of the current study and that of Jiang et al., 2022 (<https://doi.org/10.1038/s41586-022-04670-9>). The question is: what is (if any) the role of TMEM106B in disease and the potential interplay between TMEM106B and TDP-43 fibrils?

We agree that this is an important question for the field. It remains to be determined in which ways, if any, the formation of TMEM106B filaments influences neurodegenerative diseases. TMEM106B filaments have been found to accumulate in an age-dependent manner in neurologically normal individuals, as well as in those with tauopathies, synucleinopathies and TDP-43 proteinopathies (references 20-24 in the manuscript). TMEM106B inclusions do not co-localise with inclusions of tau, α -synuclein or TDP-43 (reference 23 in the manuscript).

Variation in *TMEM106B* is a risk factor for neurodegenerative disease, particularly for FTLN caused by *GRN* mutations (Van Deerlin et al. 2010. Nat. Genet. 42, 234-239). However, *TMEM106B* haplotypes do not appear to correlate with the presence of TMEM106B filaments, or influence the filament fold (references 19-24 in the manuscript). The mechanisms by which variation in *TMEM106B* contributes to neurodegenerative disease remain unknown.

Here are my specific recommendations on this important point. In lines 99-101, the authors state: “The presence of both TDP-43 filaments and TMEM106B filaments in type A FTLN-TDP, as well as in ALS with type B FTLN-TDP9, is at odds with a recent report of amyloid filaments in FTLN-TDP being composed of TMEM106B, but not TDP-43 (ref 19). Differences in the brain extraction protocol used in that study, including a lower centrifugation speed used to pellet the material examined, may have been the reason TDP-43 filaments were not observed.” However, if this was the case, then they should not see non-fibrillar TDP-43 aggregates as it is expected that non-fibrillar (protofibrillar or pathological oligomers) aggregates would settle or sediment at higher centrifugation speeds. Yet, Jiang et al. report that they do see these in their samples (See Fig. 1c of Jiang et al. paper).

The amorphous, aggregated material in Fig. 1c of Jiang et al. is large and of high-density compared to the filaments we and others observe. We would, therefore, expect that this amorphous material would sediment at lower centrifugation speeds than filaments.

Therefore, it would be important for the field to clarify the differences and similarities in the preparation or, potentially sample heterogeneity that may account for the differences.

Some specific questions/suggestions on this point are the following:

a. Is there any additional reasoning the authors could provide to support why fibrils are not detected by Jiang et al., 2022, but are in the current study?

The tissue extraction protocol of Jiang et al. is initially similar to our protocol- tissue is homogenised and incubated with 2% sarkosyl. However, Jiang et al. then centrifuge the homogenates at 21,000 g and take the pellet, whereas we centrifuge the homogenates at 27,000 g and take the supernatants. Therefore, Jiang et al. examine the opposite brain fraction to us. This is sufficient to explain why Jiang et al. did not detect filaments. We have added the following sentence to the text (line 104) to clarify this point,

'Differences in brain extraction protocols may account for this discrepancy. We found TDP-43 filaments in the supernatant following centrifugation at 27,000 g, whereas the other study examined the pellet following centrifugation at 21,000 g.'

Furthermore, if some filaments did pellet using the protocol of Jiang et al., this protocol subsequently uses the harsh detergent SDS, which would be expected to disaggregate filaments and denature protein complexes (as observed during SDS-PAGE).

b. Authors need to clarify that their extraction procedure does not induce any TDP-43 fibril formation.

TDP-43 filaments have been observed *in situ* in human brain tissue in FTLD-TDP and in ALS using immunogold, negative-stain whole-cell EM (references 13-15 in the manuscript). In these studies, the filaments have similar morphologies and widths to the filaments we observe in our extracts. Other studies have reported finding similar filaments using related extraction procedures (references 16 and 17 in the manuscript, as well as Tarutani et al. 2022. Acta Neuropathol. 143, 613-640). This suggests that our extraction procedure is unlikely to induce TDP-43 filament formation. We have added the following sentence to the text (line 83) to make it clear that such *in situ* studies have been performed,

'... consistent with previous reports of TDP-43 filaments in situ in the brains of individuals with FTLD-TDP and ALS, as well as in brain extracts.'

We also note that, for tau and amyloid- β , the use of detergent- or non-detergent-based extraction procedures does not influence the presence and structures of filaments (Shi et al. 2021. *Acta Neuropathol.* 141, 697-708; Stern et al. 2022. *BioRxiv.* DOI 10.1101/2022.10.18.512754).

c. Do the authors observe any non-fibrillar soluble TDP-43 species (pathological oligomeric TDP-43 species) in their preparations, similar to those reported by Jiang et al? To this end, authors should show negative stain TEM images as those are missing from the figures.

We have now added to ED Fig. 1e immunogold negative-stain EM images of our preparations using an antibody against TDP-43 phosphorylated at S409 and S410, which is specific to pathological TDP-43. We observed TDP-43-immunoreactive filaments, as previously reported (references 9, 16 and 17 in the manuscript, as well as Tarutani et al. 2022. *Acta Neuropathol.* 143, 613-640). We did not observe non-fibrillar objects that were immunoreactive against this antibody. We have added the following sentence to the text (line 84),

'The identity of the filaments was confirmed using immuno-gold negative-stain EM (Extended Data Fig. 1e).'

It remains to be established if the non-fibrillar material described by Jiang et al. is related to disease. Jiang et al. show that this material binds an antibody against total TDP-43 (Proteintech 60019-2-IG). However, they do not use an antibody specific for pathological TDP-43 (such as against TDP-43 phosphorylated at S409 and S410) and did not perform a negative control using extracts from a case without TDP-43 pathology. By immunoblotting, we did not find immunoreactivity against TDP-43 phosphorylated at S409 and S410 in the equivalent brain fraction using our protocol, as shown below for the reviewer, suggesting that the material described by Jiang et al. may not be disease-associated. We note that TDP-43 forms higher-order assemblies in health (reference 12 in the manuscript).

d. The authors should show both the supernatant and pellet fractions in the immunoblots to confirm or rule out the possibility of the absence of non-fibrillar TDP-43 aggregates and also to solidify their results that type-A FTL D patients have abundant fibrillar TDP-43 species.

We have now added immunoblots showing both the supernatant and pellet fractions to ED Fig. 1c. These show that the vast majority of pathological TDP-43 (phosphorylated at S409 and S410) is in the pellet fraction. This is similar to what we observed for ALS with type B FTL D-TDP (reference 9 in the manuscript). These results suggest that disease-associated TDP-43 is filamentous.

e. Authors should perform immunoelectron microscopy (similar to what they have done in <https://doi.org/10.1007/s00401-012-1055-8>) with TDP-specific to show that fibrils can be detected in the cells of the brain sections they used for cryo-EM. This way, they can strengthen their claims of detecting TDP-43 fibrils.

As in our response to point 1b, we cite studies that performed immunoelectron microscopy for TDP-43 in brain sections from FTL D-TDP and ALS patients (references 13-15 in the manuscript), which found TDP-43-immunoreactive filaments with similar morphologies and widths to the filaments we observe in our extracts, and added the following sentence to the text (line 83),

'... consistent with previous reports of TDP-43 filaments in situ in the brains of individuals with FTL D-TDP and ALS, as well as in brain extracts.'

The same analysis using TMEM106B-specific antibodies would be extremely interesting to explore whether TMEM106B and TDP-43 fibrils may co-aggregate.

Successful antibody labelling of filaments for immunoelectron microscopy requires epitopes in the fuzzy coat flanking the ordered filament core. TMEM106B filaments lack an N-terminal fuzzy coat (the N-terminal residue is buried in the ordered filament core) and have either a short (20-residue) or absent C-terminal fuzzy coat. Therefore, it has not been possible to perform successful antibody labelling during immuno-EM of extracted TMEM106B filaments. However, we note that TMEM106B inclusions do not co-localise with inclusions of TDP-43 (reference 23 in the manuscript), which suggests that the filaments may not co-aggregate. We have added the following sentence to the text (line 107) to acknowledge this,

'TMEM106B filaments have also been observed in the brains of individuals with tauopathies and α -synucleinopathies, as well as in the brains of individuals with normal neurology. TMEM106B inclusions do not co-localise with inclusions of TDP-43, tau or α -synuclein.'

2. In (line 180, 181, page 7), the authors state that “Analysis of filaments from the brains of three individuals with type A FTLD-TDP by mass spectrometry established the presence of TDP-43 molecules that were citrullinated at R293”. Can the authors clarify whether this PTM is observed exclusively in filaments or is it also possible to detect these in the non-fibrillar TDP-43 aggregates?

We do not observe non-fibrillar TDP-43 aggregates in our extracts (see response to point 1c), suggesting that this PTM is exclusive to filaments. We cannot exclude that TDP-43 may also be citrullinated in health, as acknowledged in the text (line 204),

'Possible roles for the citrullination and methylation of TDP-43 at R293 in other neurodegenerative conditions, as well as in health, remain to be determined.'

3. In lines 86-87, the authors state that “The filaments comprised a single amyloid protofilament of stacked TDP-43 molecules.” How does this explain the extra density viewed beyond R272? Is it possible that they are also appearing due to parallelly aligned protofilament? I note that data showing that the RRM fragment fits this density better than an adjacent fibril will clarify the point.

Fig. 1a shows that this extra protein density is connected to the density for R272 and is, therefore, part of the same protofilament. Below, we have included an illustration for the

reviewer showing a tentative alignment of how the 17 residues preceding R272 could be accommodated into this extra protein density. Residues from RRM2 are coloured in yellow.

4. In lines 94-96, the authors state that: “The presence of TMEM106B was confirmed using mass spectrometry, which identified peptides mapping to the C-terminal region that forms the ordered core of the filaments (Extended Data Fig. 3b)19–21.” The authors should clarify whether the mass spectrometry analysis of the TMEM106B fibrils was performed from tissues of all 3 individuals.

Originally mass spectrometry analysis of TMEM06B was only performed for individual 2. We have now added mass spectrometry analysis of TMEM106B for individuals 1 and 3. TMEM106B peptides were detected for all individuals. All peptides mapped to the region of TMEM106B that forms the ordered filament core, with the exception of one peptide from individual 2, which mapped to the transmembrane helix of TMEM106B. We have replaced the peptide table in ED Fig. 3b with a schematic to summarise these expanded results and have moved the peptide table to Supplementary Table 1.

5. In line 105, the authors claim that “There are no clear relationships between TMEM106B filament folds and diseases.” While I think that the authors refer to other studies suggesting that TMEM106B fibrils occur in an age-dependent manner in the brains of individuals that did not suffer from neurodegenerative diseases,

This sentence refers to the observation that multiple TMEM106B filament folds are present in a single disease and that the same filament fold can be found in different diseases. This is in contrast to filaments of tau, α -synuclein and TDP-43, which display a single, distinct filament fold per disease. We have updated the text (line 109) to clarify this point,

'Unlike for these proteins, there are no clear relationships between different TMEM106B filament folds and diseases.'

one cannot exclude the possibility that TMEM106B and TDP-43 interact in a pathologic manner in patient brains, perhaps even co-aggregating. This possibility should be discussed or experimentally tested by the authors.

As in our response to point 1e, we note that TMEM106B inclusions do not co-localise with inclusions of TDP-43 (reference 23 in the manuscript), which suggests that the filaments do not co-aggregate, and have added the following sentence to the text (line 107),

'TMEM106B filaments have also been observed in the brains of individuals with tauopathies and α -synucleinopathies, as well as in the brains of individuals with normal neurology. TMEM106B inclusions do not co-localise with inclusions of TDP-43, tau or α -synuclein.'

6. In lines 111-112 the authors state: “The fold consists of five connected layers formed by R272–Q360 in the LCD, with flanking regions forming the fuzzy coat.” Can the electron densities beyond R272 be fit to the residues of LCD which could contribute to the fuzzy coat? As in our response to point 3, the 17 residues preceding R272 (V255–E271) can be accommodated in this extra density. The reviewer is, therefore, correct in stating that these residues would not form the fuzzy coat. We thank the reviewer for pointing out this error and have corrected the text relating to this point (line 139), which now reads,

'Residues N-terminal to V255 and C-terminal to Q360, therefore, form the fuzzy coat of the filaments.'

7. In lines 133-134, the authors claim that “This [less ordered protein] density could accommodate approximately 18 residues (S255–E271), which would include b5 of the second

TDP-43 RRM”. It would be interesting to see modeling simulation data for this prediction or at least data predicting the propensity of this region to fold into an amyloid-like fold.

Please see our response to point 3, where we have included, for the reviewer, a model of how the 17 residues preceding R272 could be accommodated into this extra protein density. There was a typo in the manuscript- S255 should be V255. We have now corrected this.

8. In lines 134-136, the authors state that “The incorporation of CTFs that terminate between S255–E271 into the filaments may explain why this density was less ordered²⁷. Their incorporation would lead to the exposure of the hydrophobic patch.” Shouldn’t the incorporation of the residues from S255-S271 lead to the burial, rather than exposure of the hydrophobic patch? Maybe some structure prediction data of these residues aligning next to the patch could clarify the point.

As the reviewer states, the incorporation of V255-E271 would lead to the burial of the hydrophobic patch. Therefore, the incorporation of CTFs that lack all or some of V255-E271 would expose the hydrophobic patch. We have rephrased the sentence relating to this point (line 140) for clarity, which now reads,

‘The incorporation of CTFs that lack all or some of V255–E271 into the filaments may explain why this additional protein density was less ordered. Their incorporation would lead to the exposure of the hydrophobic patch.’

9. In lines 139-141, the authors conclude that “An additional peptide-like density is located on the outside surface of the 139 fifth layer, adjacent to G351–N355 (Fig. 1a,d). Its disconnected nature precluded sequence assignment. It may originate from the N- or C-terminal flanking regions, or from a separate interacting protein.” It would be very interesting to check if residues from TMEM106B could fit here.

Unfortunately, the peptide is not of sufficient length or sufficiently resolved to identify the protein by its sequence, similar to disconnected peptides in α -synuclein filament structures (reference 30 in the manuscript). We have, therefore, avoided speculation as to its identity. As in our response to point 1e, we note that TMEM106B inclusions do not co-localise with inclusions of TDP-43 (reference 23 in the manuscript), which suggest that this peptide may not be derived from TMEM106B.

10. In line 215 the authors state that “Residues A321–Q331 in the hydrophobic region form the nexus of both filament folds”. TDP-43 linked mutations are also found on residues A321 and Q331. The authors should comment on the possible effect of these mutations on the structure of the core of the filaments.

In the type A FTLD-TDP fold, A321 is part of a β -strand and faces the exterior, whereas in the ALS with type B FTLD-TDP fold it is part of a turn. TDP-43 with the A321G mutation would be able to adopt both of these folds. On the contrary, Q331 is buried in both folds and Q331K could not be accommodated in either structure. We have added a Supplementary Table 2 detailing the compatibility of the 24 mutations located within the region that forms the ordered filament core with the two filament folds. We have also added the following paragraph to the text (line 240),

'Twenty-four disease-associated TARDBP mutations are located within the region that forms the filament folds of type A FTLD-TDP and ALS with type B FTLD-TDP. Most of these mutations are compatible with at least one of the filament folds, whereas four mutations are incompatible with both folds (Supplementary Table 2). Given the observed structural variation of the type A FTLD-TDP filament fold, we do not exclude the possibility that these mutations could be accommodated in these folds by additional alternative local conformations. It is also possible that individuals with these mutations have different filament folds.'

Minor points:

1. Authors should comment on the contribution of different cell types or brain regions in driving different amyloid filament folds.

We agree that this is an interesting point for discussion. As stated on line 22, the different types of FTLD-TDP are classified based on the neuronal distribution and subcellular localisation of assembled TDP-43. We agree with the reviewer that these different cellular and subcellular localisations may favour the different amyloid filament folds. We have added the following sentence to the text (line 251) to address this point,

'The different types of FTLD-TDP are distinguished based on the neuronal distribution and subcellular localisation of assembled TDP-43, which suggests that cellular environment may affect TDP-43 filament folds.'

2. Mass spec data belongs to individuals 2, 4, 5. Since the structures obtained are mainly from individuals 1,2,3 it would be helpful to add mass spectrometry data from individuals 1 and 3. We have now carried out mass spectrometry analysis for individuals 1 and 3. We detected citrullination and mono-methylation of R293 for individual 1 and have added these results to ED Fig. 7 and 8. For individual 3, we obtained a few, poor-quality spectra for peptides containing citrullinated R293 and no peptides containing methylated R293. This is due to low peptide detection for this individual (shown in the table below for the reviewer), possibly because the tissue sample was of lower quality.

Individual	# peptides containing R293
1	90
2	178
3	22
4	54
5	83

3. On line 49 the authors state referring to: “They bind the dye thioflavin-T poorly¹⁸”. This should be thioflavin-S, not thioflavin-T.

We thank the referee for pointing out this mistake and have corrected it in the revised text. We have also incorporating the reviewer's term 'amyloidophilic dye,' which we like. The revised text (line 49) now reads,

'They bind the amyloidophilic dye thioflavin-S poorly.'

4. The authors should indicate which part of the frontal cortex (is it the motor cortex or prefrontal cortex) was used for brain extraction.

We used prefrontal cortex for all individuals and have updated the text accordingly.

Referee #2 (Remarks to the Author):

Nature 2023-02-02272

TDP-43 forms amyloid filaments with a distinct fold in type A FTLD-TDP

Arseni et al.

I would say at the outset that my expertise is in cryoEM and structural biology, not in FTLD or TDP-43-associated disorders.

The manuscript describes ex vivo structures of TDP-43 amyloid fibrils from the brains of three individuals with frontotemporal lobar degeneration with TDP-43 pathology, but who do not have accompanying ALS pathology. I have very few issues with the work as presented. This is a concise and succinct, well-written manuscript with generally clearly presented results and conclusions. The mosaicity of post-translationally modified subunits along individual fibrils is fascinating for example. I do have some specific queries and suggestions that follow, but none of them should preclude publication of the work. However, I am not convinced that the study generates sufficient new biological or medical insight to merit publication in Nature.

The fundamental insight reported is that a protein adopts a different fold in amyloid aggregates in the brains of individuals with different pathologies. However, this now seems to me to be well established – in large part in the pages of Nature, where pathology specific ex vivo structures have already been reported for Tau, A β and α -synuclein for example. So, my reaction to this well executed study on TDP-43 was: yes, I assumed this might well be the case. The study therefore feels fundamentally incremental rather field-shifting.

I offer some constructive criticism that I hope will help improve the work regardless of where it is published.

1. On L111, the authors state the fibril has a right-handed twist, and cite Fig1b-d, ED Fig 2d-e. However, these figures speak to the quality of the map/model, not its handedness. I agree that the handedness of the structure should just about be discernible at $\sim 2.4\text{\AA}$, but there is no evidence of this that I can see in any figure or methods section.

Evidence of the handedness is presented in ED Fig. 2e, which shows well-resolved side chain densities and main chain oxygen atom densities in β -strands. The sequential side chains and main chain oxygens lie on a right-handed helix. We have revised the corresponding figure legend, the end of which now reads,

'...and main chain oxygen atoms in β -strands, which reveal the chirality of the map...'

2. I was interested in the non-protein densities in the structure (Fig1d) – do these span multiple amyloid layers? – note I use layer in the 4.8Å spacing along the fibril helical axis, not the ‘five layers within the fold’ sense the authors use. It’s impossible to see this from the figures/text. We agree that this is an interesting point. The additional peptide-like densities and putative solvent molecules are separated by 4.8 Å along the filament axis, whereas larger non-protein densities within cavities appear partly contiguous. This may be because these non-protein molecules do not follow the same helical symmetry as the TDP-43 molecules. We have now added a side view of the non-protein densities to ED Fig. 4h to illustrate this, as well as the following sentence to the text (line 149),

'Unlike the protein densities, they appeared partly contiguous along the helical axis (ED Fig. 4g), possibly because they do not follow the same helical symmetry as TDP-43.'

Indeed the only side-view of the fibril I found was in ED data where the threshold used was quite generous – not allowing the reader to see clear layer separation as I expect there is at this resolution

Individual TDP-43 molecules are well separate along the helical axis at the threshold used in ED Fig. 2c and 5d. We appreciate that this is difficult to see, because the multi-layered fold is not planar along the helical axis, but instead rises and falls, as shown in ED Fig. 4b. We have now added to ED Fig. 4g a view along the helical axis limited to one layer of the fold for five TDP-43 molecules, where the separation of TDP-43 molecules is more apparent.

3. Indeed, I found the ‘five-layer’ description unhelpful in a structural field where ‘layer’ normally refers to the 4.8Å spacing along the fibril access. This was perhaps more so because it’s not really used or explained in their figures – if the authors feel this terminology is useful, I would suggest it is established in Figure 1.

We would like to retain our use of the term 'layer' for consistency with published studies of amyloid filaments (Zhang et al. 2020. Nature 580, 283-287; Schweighauser et al. 2020. Nature 585, 464-469; Shi et al. 2021. Nature 598, 359-363; Schweighauser et al. 2022. Nature 605, 310-314; Yang et al. 2022. Nature 610, pages 791–795). However, we agree with the reviewer that this term could be better explained in the manuscript. We tried to indicate the layers of the filament fold by colouring them in ED Fig. 4a. We have now used different colours for each layer to make this clearer in this figure. As suggested by the reviewer, we have also labelled the layers in Fig. 1d.

4. The alternate local conformations in Figure 2 are very cool – as is the MS backup to show the PTMs involved.

5.

Minor Points.

1. Personally I find it very helpful for the data availability statement to directly link which Accession code relates to which structure

We agree and have now detailed which accession codes relate to which structures in the data availability statement.

2. Do the authors intend to deposit the raw micrograph movies in EMPIAR? There is no reference to this in the data accessibility statement If not why not? Yes, this was our intention. We have now completed the deposition of the raw micrograph movies to EMPIAR and have added the accession codes to the data availability statement.

Referee #3 (Remarks to the Author):

In this paper, Ryskeldi-Falcon and coworkers present cryo-EM structures of TDP-43 fibrils isolated from type A FTLD-TDP. Remarkably, these fibrils adopt a different conformation than TDP-43 fibrils formed in ALS and type B FTLD-TDP. They also find TMEM106B fibrils in type A FTLD-TDP patient extracts, which are anticipated from prior studies. Intriguingly, they find a buried arginine (R293) in the fibril core, which might be predicted to be destabilizing. However, they also find that R293 is likely citrullinated in assembled fibrils, which may explain

how they form. They also find that in some TDP-43 molecules, R293 can be monomethylated, but this modification is likely incompatible with the fibril core, which is consistent with in vitro findings that monomethylation of TDP-43 at R293 can reduce TDP-43 aggregation. Overall, these studies are a tour de force and establish the formation of distinct TDP-43 fibril ‘strains’ in distinct neurodegenerative disorders. They set the stage for therapeutic strategies to target the distinct forms of assembled TDP-43 in distinct disorders. I recommend publication without delay.

Reviewer Reports on the First Revision:

Referees' comments:

Referee #1 (Remarks to the Author):

I now had the chance to review the revised manuscript of Arseni et al and I applaud the authors for a fantastic job that improved their manuscript by addressing all the points of criticism of the referees. I recommend publication of the work in Nature without further delay.

The work strongly supports the notion of “TDP-43 strains”, meaning that the different fibrillar structures trigger different toxic pathways in patients with different disease subtypes. In my view, this constitutes a major discovery that will be influential in the ALS/FTD field, even if similar data exist for other neurodegenerative diseases.

Referee #2 (Remarks to the Author):

Firstly, I'd like to apologise to the authors and editorial team for the delay in submitting this review. It's been a busy time!

I'd like to thank the authors for a careful and thoughtful response. The changes that they have made in response to the referee comments (both mine and particularly referee #1) have improved the manuscript, which is now even more solid.

However, as I hope I made clear in the first review, I had few scientific criticisms of the structural work as originally presented. The publication decision is essentially an editorial one, that I do not wish to obstruct. My view remains that this very strong paper does not really present a fundamental advance that provides a step change in our understanding in this field. If Nature's view is that it does, then they should be confident that the work is clear, concise and scientifically solid.